# Idiotype-specific CD4+ T cells chronically stimulate autoreactive B cells to develop into B lymphomas in mice

Ramakrishna Prabhu Gopalakrishnan [1,2] ✉, Jerrold M. Ward [3],
Victor Greiff [1,2], Ranveig Braathen [1,2], Xian Hu[4], Livia Bajelan[1], Khang Lê Quý[1,2],
Ludvig Munthe [1,5], Peter Csaba Huszthy [2,6] & Bjarne Bogen [1,2] ✉

Autoimmunity increases the risk of developing B lymphoma in humans; however, the molecular mechanism(s) underlying this link remain(s) unexplained. Here, we develop a mouse model to dissect the contribution of Id-driven T-B collaboration and show that chronic interaction between B cells and CD4+ T cells first leads to autoimmunity and later to the development of B (and T) cell lymphomas. We find that serum autoantibodies and lymphoma B cell receptor (BCR) are related and have the same specificity for ubiquitous self-antigens (histone and nucleosome) (Signal 1). Self-reactive B lymphoma cells are helped by CD4+ T cells that recognize a lymphoma neoantigen, an MHC class II-presented Idiotypic (Id) peptide (Signal 2). The mechanism, called Id-driven T-B collaboration, results in relentless mutual stimulation of B and T cells with germinal center markers, autoimmunity, and finally, malignant transformation of either B or T cells. Our results thus indicate Id-driven T-B collaboration as a potential mechanism linking autoimmunity and the development of lymphomas.

Several different mechanisms have been described that contribute to the development of autoimmune diseases[1,2] and B-cell lymphomas[3,4]. However, in most cases, the etiologies of these heterogeneous diseases are unknown. Autoimmune diseases and B lymphomas are linked disorders in humans[5–7]. However, the mechanism(s) underling the joint pathogenesis of these diseases have not been fully elucidated.

Due to somatic hypermutation in germinal centers[8], B cells often express mutated sequences [neoantigens, traditionally called idiotypic (Id) peptides] in their B cell receptor (BCR) variable (V)-regions. Moreover, B cells have been shown to antigen-process their own BCR, resulting in Id-peptides that are presented on their MHC class II molecules to Id-specific CD4+ T cells[9]. This phenomenon is the basis for the concept of Id-driven T-B collaboration[10] which is distinct from conventional T-B collaboration[11,12]. A growing body of evidence suggests that Id-driven T-B collaboration may cause the development of autoimmunity and B-cell lymphoma. Studies in paired Ig and TCR-transgenic mice have demonstrated that chronic Id-driven T-B collaboration can cause the development of autoimmune disease with SLE-like features[12–14]. Moreover, injection of Id-specific Th2 cells into mice that are transgenic for an Id+ L chain resulted in the development of marginal zone B lymphomas[15].

These previous findings in gene-modified mouse models are supported by observations in humans. Thus, Id-specific CD4+ T cells have been observed in patients with SLE[16], rheumatoid arthritis[17], and multiple sclerosis[18]. Moreover, Id-specific CD4+ T cells stimulated B cell Chronic Lymphocytic Leukemia (B-CLL) in vitro[19]. Finally, MHCII-

[1]Department of Immunology and Transfusion Medicine, Division of Laboratory Medicine, Oslo University Hospital, Oslo, Norway. [2]Department of Immunology, Institute of Clinical Medicine, University of Oslo, Oslo, Norway. [3]National Cancer Institute, NIH, Bethesda, MD, USA. [4]Center for Cancer Cell Reprogramming, Faculty of Medicine, University of Oslo, Oslo, Norway. [5]KG Jebsen Centre for B cell Malignancies and Precision Immunotherapy Alliance, University of Oslo, Oslo, Norway. [6]Department of Microbiology and Infection Control, Akershus University Hospital, Lørenskog, Norway. ✉e-mail: r.p.gopalakrishnan@medisin.uio.no; bjarne.bogen@medisin.uio.no

presented Id-peptide neoantigens were frequently found in mantle cell lymphoma[20], diffuse large B cell lymphoma (DLBCL), follicular B cell lymphoma (FBL) and B-CLL[21].

Evidently, Id-driven T-B collaboration must be tightly controlled by a number of mechanisms to prevent rampant lymphocyte activation and immunological havoc in normal individuals. First, T cells are tolerant to germline-encoded V region sequences and only respond to Id-peptide neoepitopes expressing somatic mutations[22–24]. Second, resting normal B cells display only low levels of pId:MHCII complexes on their cell surface[25,26]. Third, Id-driven T-B collaboration is likely to be suppressed by mechanisms that remain to be elucidated.

In previous studies on Id-driven T-B collaboration, we used $\lambda2^{315}$ Ig L chain transgenic mice in which most B cells present a mutated $Id^{315}$ peptide on their MHC class II molecules[13,15]. To obtain a more physiological model, we recently developed a V gene segment-modified mouse ($V\lambda2^{315m}$) where a $V\lambda2^{315m}$-J$\lambda2$ rearrangement in infrequent B cells generates a complete Id-sequence spanning the V-J junction[26]. Importantly, ligation of the BCR by antigen resulted in increased display of pId:MHCII on the B cell surface and hence elicitation of Id-driven T-B collaboration[26].

Here, we hypothesize that chronic Id-driven T-B collaboration may explain the known link between autoimmunity and B lymphomas (Fig. 1A). We show that BCR ligation by self-antigen (Signal 1) and chronic help from Id-specific CD4$^+$ T cells (Signal 2) result in autoimmunity and later development of B cell lymphomas. These findings may have implications for the treatment of B lymphomas in humans.

## Results

### Incomplete T cell tolerance in mice that co-expressed the Id-modified V gene segment and the Id-specific TCR

In the $V\lambda2^{315m}$ mouse, germline codons in the $V\lambda2$ gene segment were replaced by the corresponding mutated codons found in the MOPC315 (IgA$\lambda2$) multiple myeloma cell line, resulting in a Tyr$^{94}$Ser$^{95}$Thr$^{96}$→Phe$^{94}$Arg$^{95}$Asn$^{96}$ exchange. In heterozygous $V\lambda2^{315m+/-}$ mice, about 0.5% of B cells rearrange the $V\lambda2^{315m}$ to J$\lambda2$; these B cells express a $\lambda2$ CDR3 Id-peptide (residues 91–101) that spans the $V\lambda2^{315m}$J$\lambda2$ junction. This Id-peptide neoantigen, when bound to the MHC class II molecule I-E$^d$ of BALB/c mice, stimulates Id-specific CD4$^+$ T cells[26]. The $V\lambda2^{315m}$ mice are close to physiological since their B cell development appears normal and since the Id-sequence is expressed at physiological levels in a small population of B cells[26].

Heterozygous $V\lambda2^{315m}$ mice were crossed with heterozygous Id-specific TCR transgenic (Tg46) mice[27], yielding four types of offspring on a BALB/c background (Fig. 1B). According to the hypothesis (Fig. 1A), the $V\lambda2^{315m+/-}$TCR-TG$^{+/-}$ offspring could develop autoimmunity and perhaps even B cell lymphomas. The other three types of offspring, singly V gene-modified, singly TCR-transgenic and non-transgenic, served as experimental controls.

We first tested if T cell tolerance developed in $V\lambda2^{315m+/-}$ TCR-TG$^{+/-}$ offspring. In 4-week-old mice, frequencies of double positive (DP) and single positive (SP) thymocyte subsets (gated as shown in Supplementary Fig. 1A) were unaffected (Supplementary Fig. 2A). Nevertheless, $V\lambda2^{315m+/-}$ TCR-TG$^{+/-}$ offspring had a reduction in the frequency of Id-specific T cells among CD4+ SP thymocytes compared to TCR-TG mice (Fig. 1C, and Supplementary Fig. 2B). Similarly, $V\lambda2^{315m+/-}$ TCR-TG$^{+/-}$ offspring had a reduction of Id-specific cells among CD4+ T cells in the spleen (Fig. 1D, and Supplementary Figs. 1B, 2C). The splenic Id-specific CD4+ T cells that remained had an increased frequency of FoxP3+ cells, suggestive of increased differentiation towards a regulatory phenotype (Fig. 1E, and Supplementary Fig. 2D). Despite these signs of partial tolerance, the proliferative responses of Id-specific CD4+ T cells isolated from $V\lambda2^{315m+/-}$ TCR-TG$^{+/-}$ mice to synthetic Id-peptide in vitro were undiminished (Fig. 1F). These results indicate that Id-specific CD4+ T cells became partly but not completely tolerized in the presence of about 0.5% of Id+ B cells. B cell subsets in bone marrow (BM) and spleen were similar in $V\lambda2^{315m+/-}$ TCR-TG$^{+/-}$ and $V\lambda2^{315m+/-}$ (Supplementary Figs. 1C, D and 2E, F).

### Development of autoimmunity in mice that co-expressed the Id-modified V gene segment and the Id-specific TCR

$V\lambda2^{315m+/-}$TCR-TG$^{+/-}$ offspring had a slightly reduced gain of body weight compared to littermates. Growth retardation was slightly more prominent in females than males (Fig. 1G, and Supplementary Fig. 2G). Some mice developed inflamed eyelids and moderate hair loss but appeared otherwise healthy.

With age, ~70% $V\lambda2^{315m+/-}$TCR-TG$^{+/-}$ offspring developed autoantibodies detected by staining of HEp-2 cells. Autoantibodies were predominantly antinuclear antibodies (ANA) that gave a homogeneous staining pattern (Fig. 1H). Autoantibody titers were low on day 100 but increased by day 300 (Fig. 1I, and Supplementary Fig. 2H). IgG2a autoantibodies were more prominent than IgG1 (Fig. 1I). Surprisingly, the autoantibodies expressed κ but not λ L chains (Fig. 1J). Autoantibody levels were higher in females than in males (Fig. 1K). Autoantibodies bound nucleosomes (Fig. 1L, M) and histones (Fig. 1N, O); in either case, IgG2a dominated. Levels of anti-histone and anti-nucleosome IgG antibodies correlated (Fig. 1P). Only 2/15 mice had anti-dsDNA antibodies of the IgG2a isotype in ELISA; these two sera were also positive in a *Crithidia Luciliae* assay (CLIFT assay) (Supplementary Fig. 2I, J).

### Id-specific T cells and Id$^+$ B cells proliferated in autoimmune $V\lambda2^{315m+/-}$TCR-TG$^{+/-}$ mice

At 11 months of age, $V\lambda2^{315m+/-}$TCR-TG$^{+/-}$ mice without overt lymphoma had a population of peripheral blood B cells whose BCRs had a strikingly reduced κ expression. These κ$^{low}$ B cells expressed increased amounts of MHCII, pId$^{315}$:I-E$^d$ [detected by a scFv TCRmimetic (TCRm)[26]] and CD86 (Supplementary Fig. 2K–P), indicating an expansion of B cells that expressed $\lambda2^{315m}$. Such κ$^{low}$TCRm$^+$CD86$^{Hi}$ B cells were neither observed in blood of age-matched controls nor in young (2-months-old) $V\lambda2^{315m+/-}$TCR-TG$^{+/-}$ mice (Supplementary Fig. 2Q).

At 12 months of age, splenic Id-specific CD4$^+$ T cells had expanded in autoimmune $V\lambda2^{315m+/-}$TCR-TG$^{+/-}$ mice and attained the same frequency as in TCR-TG$^{+/-}$ mice (Supplementary Fig. 3A). To more stringently test for proliferation of Id-specific CD4$^+$ T cells, 12-month-old mice without overt lymphoma received BrdU for 7 days before analysis of spleens. CD4$^+$ T cells incorporated more BrdU (proliferated) in $V\lambda2^{315m+/-}$TCR-TG$^{+/-}$ mice compared to TCR-TG$^{+/-}$ mice (Fig. 2A). Moreover, Id-specific CD4$^+$ T cells stained by a clonotype-specific mAb proliferated more extensively than did non-specific CD4$^+$ T cells (which expressed endogenous TCRα chains[27]) (Fig. 2A, and Supplementary Fig. 3B). Similar findings were done for T follicular helper (Tfh) cells[28] (Fig. 2B, and Supplementary Fig. 3C, E) as well as peripheral helper T (Tph) cells[29,30] (Supplementary Fig. 3D, trending but not significant, Supplementary Fig. 3F).

In the same experiment, B cells expressing pId$^{315}$:MHCII$^+$ proliferated to a larger extent in autoimmune $V\lambda2^{315m+/-}$TCR-TG$^{+/-}$ mice compared to $V\lambda2^{315m+/-}$ mice (Fig. 2C, and Supplementary Fig. 3G). The proliferating cells expressed BCRs with κ but not λ L chains (Fig. 2C) and the GC-marker CD95 (Supplementary Fig. 3H, trending but not significant). Consistent with these results, both the pId$^{315}$:MHCII$^+$κ$^+$ and the CD95$^+$κ$^+$ B cell subsets were expanded in autoimmune mice (Supplementary Fig. 3I, J). These results suggest a mutual stimulation between Id-specific T cells and pId$^{315}$:MHCII$^+$ B cells in GCs in the spleens of autoimmune mice, consistent with ongoing Id-driven T-B collaboration.

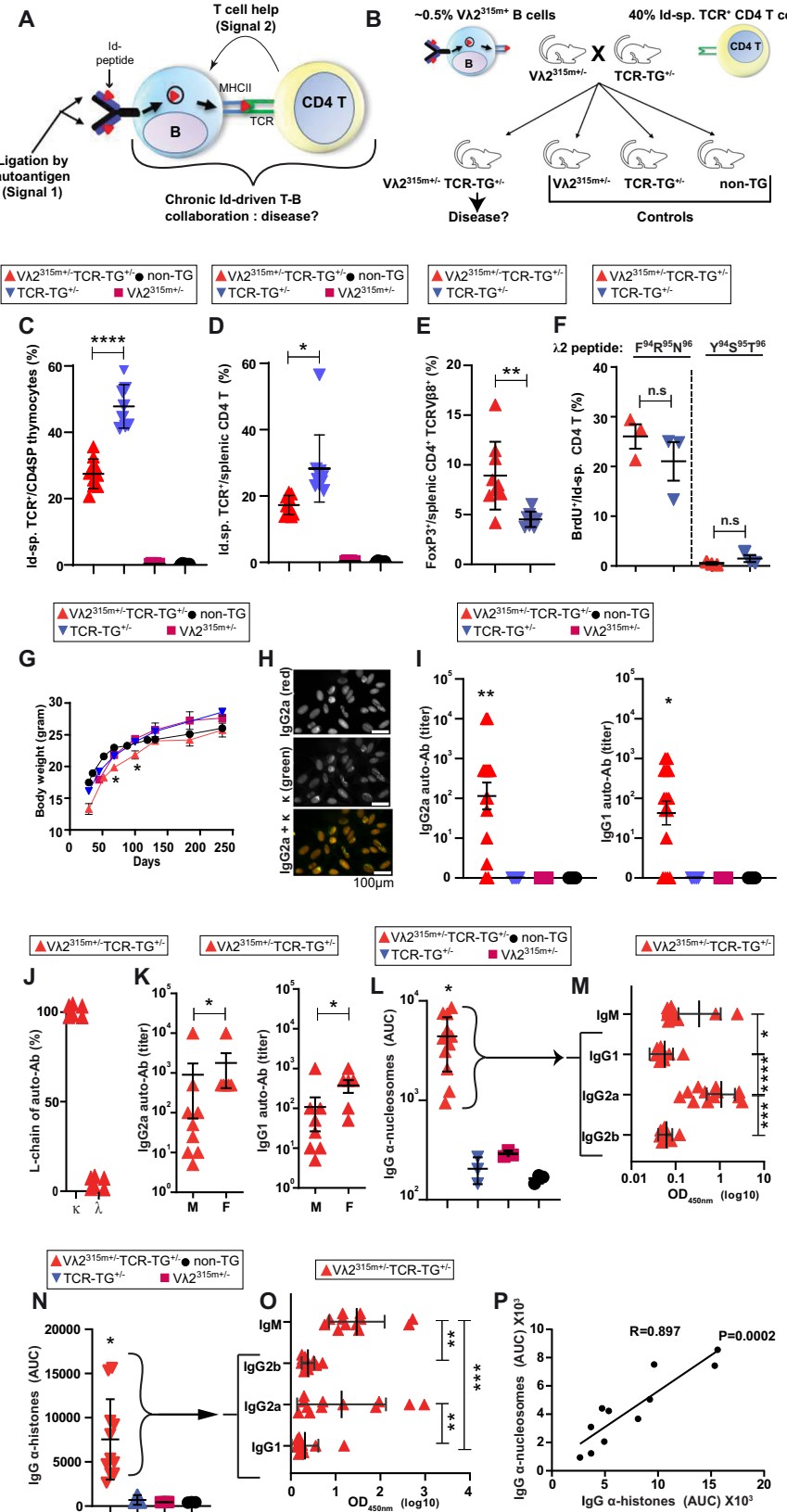

## Autoimmune mice have increased serum levels of cytokines associated with Tfh and DLBCL

We tested sera of 10-month-old mice for 23 chemokines and cytokines in a Multiplex assay. Autoimmune $V\lambda2^{315m+/-}TCR-TG^{+/-}$ mice expressed increased levels of IL-21 and IFNγ compared to controls while results with IL-10 and TNF were less pronounced (Fig. 2D–G). However, the low levels of IL-10 and TNF were strikingly increased in 14–20-month-old mice that had developed B lymphomas (see below) while levels of IL-21 and IFN-γ were stationary (Fig. 2D–G). IL-21 has previously been associated with Tfh cells[31] while IL-10 has been associated with DLBCL in humans[32].

**Fig. 1 | Hypothesis, experimental design and development of autoimmunity.**
**A** Hypothesis: can chronic Id-driven T-B collaboration model cause disease development? **B** Experimental design. Heterozygous V gene modified Vλ2$^{315m+/-}$ mice were crossed with heterozygous Id-specific TCR-TG$^{+/-}$ mice, yielding Vλ2$^{315m+/-}$ TCR-TG$^{+/-}$ mice and the indicated 3 types of littermate controls. Illustrations were made with Microsoft PowerPoint. **C–F** Characterization of Id-specific CD4$^+$ T cells in 4-week-old offspring. **C** Frequency of Id-specific TCR$^+$ cells (detected by the clonotype-specific GB113 mAb) among thymic CD4 single positive (SP) cells. $p < 0.0001$. Mean (±SD). Vλ2$^{315m+/-}$ TCR-TG$^{+/-}$, $n = 10$; TCR-TG$^{+/-}$, $n = 8$; Vλ2$^{315m+/-}$, $n = 6$; non-TG, $n = 6$. Data from three independent experiments were combined. Gating was done as shown in Supplementary Fig. 1A. **D** Id-specific TCR$^+$ cells among splenic CD4$^+$ T cells. $p = 0.0388$. Mean (±SD). Vλ2$^{315m+/-}$ TCR-TG$^{+/-}$, $n = 11$; TCR-TG$^{+/-}$, $n = 10$; Vλ2$^{315m+/-}$, $n = 6$; non-TG, $n = 6$. Data from three independent experiments were combined. Gating was done as shown in Supplementary Fig. 1B. **E** FoxP3 expression among splenic CD4$^+$ T cells expressing the transgenic TCRVβ8. $p = 0.0016$. Mean (±SD). $n = 9$/group. Pooled data from two independent experiments. **F.** In vitro proliferation (BrdU incorporation) of splenic Id-specific CD4$^+$ T cells in response to splenic APC and either the mutated Id-peptide (F$^{94}$R$^{95}$N$^{96}$) or the non-mutated (germline) version (Y$^{94}$S$^{95}$T$^{96}$). $p = 0.4$ and 0.2 for F$^{94}$R$^{95}$N$^{96}$ and Y$^{94}$S$^{95}$T$^{96}$, respectively. Mean (±SD). $n = 3$, technical replicates. One representative experiment out of two. **G** Weight development (females) of the four types of offspring. $p = 0.0015$ (day 68), $p = 0.0046$ (day 100). Mean (±SD). Mean (±SD). $n = 7$/group. **H** Serum autoantibodies detected by HEp-2 staining. Shown is IgG2a,κ antinuclear autoantibodies (ANA) in a Vλ2$^{315m+/-}$ TCR-TG$^{+/-}$ mouse on day 300. Scale bar: 100 μm. **I.** IgG2a$^+$ and IgG1$^+$ ANA in day 300 sera. $p = 0.0082$ (IgG2a$^+$) and $p = 0.0151$ (IgG1$^+$). Mean (±SEM). Vλ2$^{315m+/-}$ TCR-TG$^{+/-}$, $n = 15$; controls, $n = 6$/groups. **J** κ and λ L-chain expression in ANA in sera of Vλ2$^{315m+/-}$ TCR-TG$^{+/-}$ mice. $n = 6$. **K** Comparison of IgG2a and IgG1 ANA levels in sera of male ($n = 12$) and female ($n = 7$) Vλ2$^{315m+/-}$ TCR-TG$^{+/-}$ mice. $p = 0.0459$ (IgG2a) and 0.0493 (IgG1). **L** Anti-nucleosome antibodies in sera given as area under the curve (AUC) in ELISA. $p \le 0.0233$. Mean (±SEM). Vλ2$^{315m+/-}$ TCR-TG$^{+/-}$, $n = 12$; controls, $n = 6$/group. **M** Ig isotypes of anti-nucleosome antibodies in Vλ2$^{315m+/-}$ TCR-TG$^{+/-}$ sera ($n = 12$). $p$ values: 0.0138 (*) and <0.0001 (***). Mean (±SEM). **N.** Anti-histone antibodies in sera measured by ELISA. $p \le 0.0442$. Mean (±SEM). Vλ2$^{315m+/-}$ TCR-TG$^{+/-}$, $n = 11$; controls, $n = 4$/group. **O** Ig isotypes of anti-histone serum antibodies in Vλ2$^{315m+/-}$ TCR-TG$^{+/-}$ sera ($n = 12$). $p$ values: ≤0.0059 (**) and <0.0001 (***). **P** Correlation of anti-histone and anti-nucleosome autoantibodies in individual mice. $n = 11$. **G–P** Data from cohort one (out of three) (see Supplementary Fig. 3K for cohort information) are presented. Statistical comparisons: One-way ANOVA (Dunnett's test): **C**; Kruskal–Wallis test (two-sided), Dunn's multiple comparison: **D, I, L, M, O**. $^*p < 0.05$, $^{**}p < 0.01$, $^{***}p < 0.001$ and $^{****}p < 0.0001$. Source data are provided as a Source Data file.

## Autoimmune mice later developed lymphomas

Between 12 and 22 months of age, most Vλ2$^{315m+/-}$ TCR-TG$^{+/-}$ offspring were euthanized due to distended abdomens caused by lymphomas, while control mice were healthy (Fig. 2H). It did not influence disease development whether the mother of the offspring was Vλ2$^{315m+/-}$ or TCR-TG$^{+/-}$ (Supplementary Fig. 3K). Lymphomas developed more frequently in mice with high titers of IgG2a autoantibodies (Fig. 2I) and in females (Fig. 2J).

The lymphomas were investigated as outlined in Fig. 2K. The primary lymphomas predominantly developed in spleens and lymph nodes (Fig. 2L). Upon transfer to Rag1$^{-/-}$ mice, recipients developed secondary lymphomas. Lymphomas were classified according to the Bethesda proposals for classification of lymphoid neoplasms in mice[33]. About 60% of the lymphomas were of B cell origin. Of these, follicular B lymphoma (FBL), centroblastic diffuse large B cell lymphoma (CB-DLBCL) and histiocyte-associated (HA) DLBCL (HA-DLBCL) were of approximately equal frequency while there were fewer immunoblastic (IB)-DLBCL (Fig. 2M). Images of representative HE-stained sections are shown in Fig. 2N. 4/4 B lymphomas (2 FBL and 2 DLBCL) were BCL6$^+$, suggestive of a GC-origin[34] (Fig. 2O).

About 40% of the lymphomas were T cell lymphoblastic lymphomas (T-LBL) (Fig. 2M). Three types could be distinguished based on CD4, CD8 and TCR expression. In order of frequency, the T lymphomas were: CD4$^-$8$^-$ Id-specific TCR$^+$ > CD4$^+$8$^-$ Id-specific TCR$^+$ > CD4$^-$8$^-$ Id-specific TCR$^-$ (Supplementary Fig. 3L). The lymphomas expressed the transgenic TCR, PD-1 and CD25 at levels similar to CD4$^+$ T cells from TCR-TG mice while their CD44 expression was reduced (Supplementary Fig. 3M–P). 2/2 T-LBL (both CD4$^+$Id-specific TCR$^+$) were Bcl6$^+$ suggestive of a Tfh origin[35] (Fig. 2O). Examples of immunostaining for CD3 (T cell marker), PAX5 (B cell marker) and IBA-1 (macrophage marker) are shown in Supplementary Fig. 3Q.

## Characterization of primary B lymphomas

Primary splenic B lymphomas had increased expression of CD80, CD86, MHC class II, germinal center markers (CD95$^+$GL-7$^+$) and Ki-67 compared to B cells of control mice (Fig. 3A–E, and Supplementary Fig. 4A–E). Secondary B lymphomas in Rag 1$^{-/-}$ recipients expressed increased levels of IgG2a BCR and had downregulated IgD and IgM (Fig. 3F). Surprisingly, primary B lymphoma mice had a reduced frequency of cell surface λ$^+$CD19$^+$ cells (Fig. 3G); even so, pId$^{315}$:I-E$^d$ expression was increased (Fig. 3H, I). An explanation could be that λ2$^{315m}$ chains were only present intracellularly in B lymphoma cells but were nevertheless available for MHC class II presentation. Indeed, Western blots revealed lymphoma cell expression of λ2/3L chains in cell lysates (Fig. 3J). Further, a PCR employing Vλ2$^{315m}$-specific primers demonstrated that B lymphomas expressed Vλ2$^{315m}$ mRNA (Fig. 3K), a result which was confirmed by sequencing in 5/5 cases. Lymphomas expressed lower than normal amounts of κ on their cell surface (Fig. 3L, M), perhaps due to autoreactivity of the IgG2aκ BCR (see later).

## Dual L chain expression in a B lymphoma cell line: cell surface κ and intracellular λ

A secondary B lymphoma was adapted to growth in vitro. The cell line, NT24, expressed κ on the cell surface while λ was expressed predominantly intracellularly as determined by flow cytometry (Fig. 3N). Nevertheless, about 30% of NT24 cells expressed pId$^{315}$:I-E$^d$ on the cell surface (Fig. 3O). Moreover, the intracellular λ was identified as λ2$^{315m}$ by proteomic analysis of cell lysates precipitated by a Vλ-specific mAb (Fig. 3P). Confocal microscopy confirmed that κ was expressed on the surface while λ was expressed intracellularly (Fig. 3Q). Intracellular λ co-localized with an ER marker (Fig. 3R).

## B lymphomas were infiltrated by Id-specific CD4+ T cells that were activated and proliferated

Splenic B lymphomas were infiltrated by CD3$^+$ T cells (Fig. 4A) and intimate contacts (synapses) between B lymphoma cells and Id-specific T cells were observed in the spleen (Fig. 4B). One third of infiltrating CD4$^+$ T cells expressed the Id-specific TCR (Fig. 4C) were activated (CD69$^+$) (Fig. 4D, and Supplementary Fig. 4I), proliferated (Ki67$^+$) (Fig. 4E, and Supplementary Fig. 4J) and displayed increased levels of the memory marker CD44 (Fig. 4F, and Supplementary Fig. 4K) and PD-1 (Fig. 4G, and Supplementary Fig. 4L).

## Bidirectional activation of lymphoma B cells and Id-specific CD4$^+$ T cells

Ex vivo B lymphoma cells proliferated in response to Id-specific T cells (Fig. 4H). Conversely, B lymphomas induced proliferation of Id-specific T cells (Fig. 4I). When B lymphoma cells and infiltrating T cells were isolated from a single tumor and mixed in vitro, B lymphoma cells enhanced the proliferation of Id-specific T cells demonstrating that TILs were not fully anergic (Fig. 4J). These results were extended to a cloned B lymphoma cell line, NT34.E2 (Supplementary Fig. 4F–H), which similarly stimulated the proliferation of Id-specific CD4$^+$ T cells in vitro (Fig. 4K).

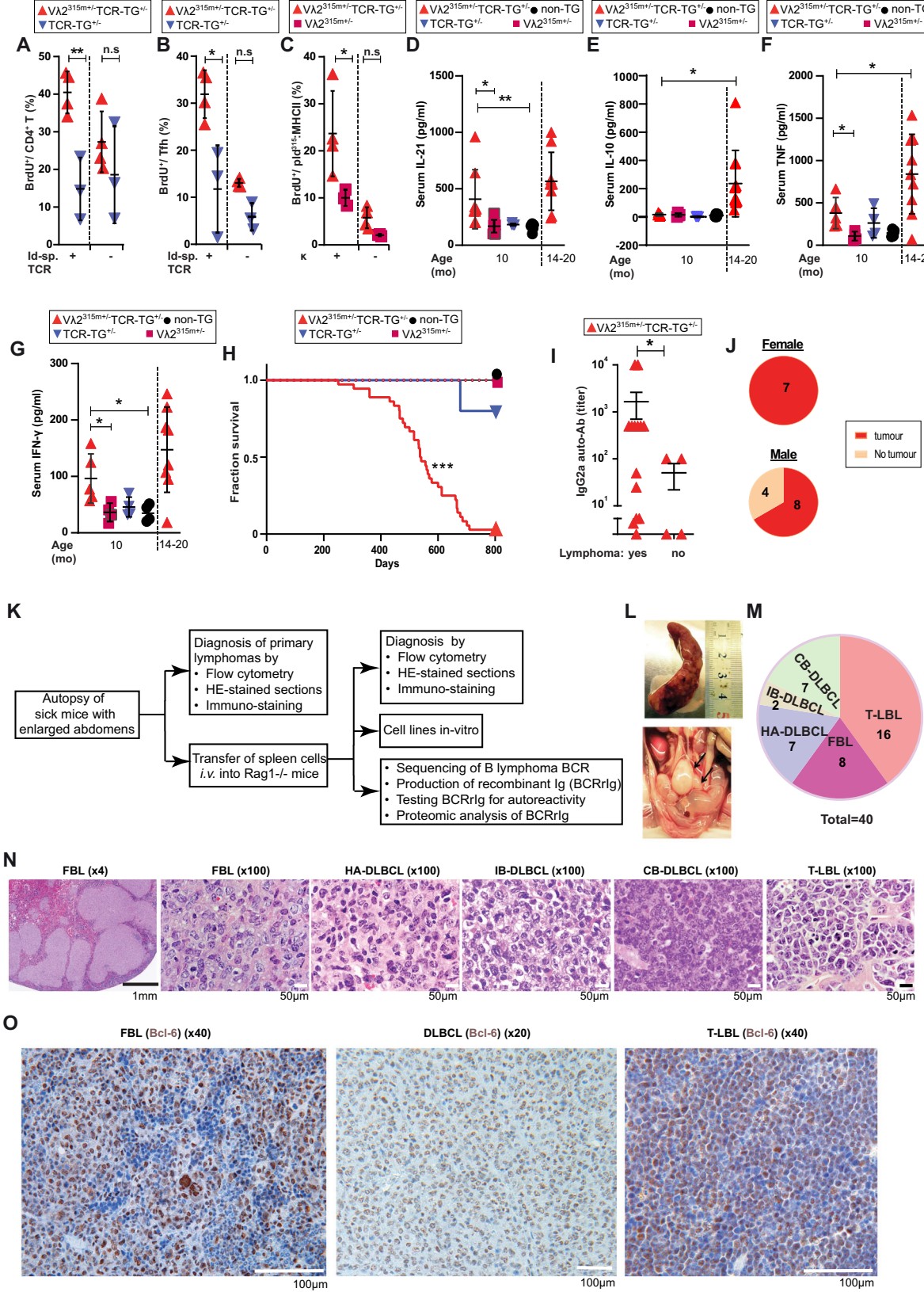

## B lymphoma growth was retarded in Id-specific TCR-transgenic mice

NT34 cells were labeled with luciferase and injected into Id-specific TCR-transgenic Rag1$^{-/-}$ mice and Rag1$^{-/-}$ controls. In vivo imaging showed that B lymphoma growth was retarded in mice with Id-specific CD4$^+$ T cells (Fig. 4L and Supplementary Fig. 4M). Consistent with this,

survival was prolonged in challenged TCR-transgenic mice although mice eventually succumbed to disease (Fig. 4M).

## Characterization of lymphoma BCRs

Four independent IgG2a, κ B lymphomas (NT7, NT8, NT18, NT24) were passaged in Rag1$^{-/-}$ mice. mRNA isolated from secondary lymphomas

**Fig. 2 | Lymphoma development. A–C** 12-month-old Vλ2[315m +/−] TCR-TG[+/−] mice without overt lymphoma disease and age-matched controls received BrdU until analysis of spleens on day 7. **A** Frequency of BrdU[+] cells among CD4[+] T cells in Vλ2[315m+/−] TCR-TG[+/−] compared to TCR-TG[+/−] mice. Id-specific and non-Id-specific CD4[+] T cells were analyzed. $p = 0.0033$ and $0.2212$. **B** Frequency of BrdU[+] cells among gated Tfh (CXCR5[+]PD-1[+]) cells. $p = 0.0286$ and $0.3198$. **C** Frequency of BrdU[+] cells among B cells that displayed pId[315]:MHCII(I-E[d]), detected by a TCR mimetic scFv(TCRm). Vλ2[315m +/−] TCR-TG[+/−] were compared to Vλ2[315m+/−] mice. κ[+] and κ[-] B cells were analyzed. $p = 0.0273$ and $0.0571$. Mean (±SD). Vλ2[315m +/−] TCR-TG[+/−], $n = 4$; TCR-TG[+/−], $n = 3$; Vλ2[315m+/−], $n = 3$. One representative experiment out of two independent experiments is presented. **D–G** Cytokine measurement in sera from respectively 10-month-old and 14–20-month-old Vλ2[315m+/−] TCR-TG[+/−] mice, compared to control mice. **D** IL-21. $p = 0.0433$ and $0.0037$. Mean (±SD). 10-months-old mice: Vλ2[315m+/−] TCR-TG[+/−], $n = 9$; Vλ2[315m+/−], $n = 10$; TCR-TG[+/−], $n = 8$; non-TG, $n = 9$. 14–20-months-old mice: Vλ2[315m+/−] TCR-TG[+/−], $n = 9$. **E.** IL-10. $p = 0.0275$. Mean (±SD). 10-months-old mice: Vλ2[315m+/−] TCR-TG[+/−], $n = 7$; Vλ2[315m+/−], $n = 4$; TCR-TG[+/−], $n = 4$; non-TG, $n = 9$. 14–20-months-old mice: Vλ2[315m+/−] TCR-TG[+/−], $n = 9$. **F** TNF. $p = 0.0477$. Mean (±SD). **G** IFN-γ. $p = 0.0299$ and $0.0268$. Mean (±SD). In both **F** and **G**, 10-months-old mice: Vλ2[315m+/−] TCR-TG[+/−], $n = 5$; Vλ2[315m+/−], $n = 4$; TCR-TG[+/−], $n = 4$; non-TG, $n = 4$. 14–20-months-old mice: Vλ2[315m+/−] TCR-TG[+/−], $n = 9$. **H** Survival curves of Vλ2[315m+/−] TCR-TG[+/−] mice compared to control mice. $p = 0.0001$. Vλ2[315m+/−] TCR-TG[+/−], $n = 40$; Vλ2[315m+/−], $n = 19$; TCR-TG[+/−], $n = 14$; non-TG, $n = 19$. **I** Levels of IgG2a ANA in 10-month-old mice that later either developed lymphomas or not. $p = 0.047$. Mean (±SEM). Lymphoma group, $n = 14$; No lymphoma group, $n = 4$. **J** Incidence of lymphomas in female and male mice. Female, $n = 7$; male, $n = 12$. **K** Flowchart of lymphoma analyses. **L** Photomicrographs of a spleen (top) and lumbar lymph nodes (bottom) in a representative mouse with B lymphoma. Enlarged lymph nodes are indicated by arrows. **M** Classification of lymphoma types based on immunohistology. T-LBL: T cell lymphoblastic lymphoma, FBL: Follicular B lymphoma, HA-DLBCL: Histocyte associated DLBCL (Diffuse Large B Cell Lymphoma), IB-DLBCL: Immunoblastic DLBCL, CB-DLBCL: Centroblastic DLBCL. $n = 40$ lymphomas. **N** HE-staining of the indicated types of lymphomas. Scale bar: 1 mm and 50 μm for 4X and 100X, respectively. **O** Bcl-6 staining (in brown) of the indicated types of lymphomas. Scale bar: 100 μm. **N, O** One representative example of each lymphoma shown out of 40 analyzed (in **M**). Statistical comparisons: Mann–Whitney U test (two-tailed): **A–C, I**; Kruskal–Wallis test (two-sided), Dunn's multiple comparison: **D–G**; Mantel–Cox test: **H**. *$p < 0.05$, **$p < 0.01$ and ***$p < 0.001$. Source data are provided as a Source Data file.

were used to amplify VDJ$_H$ and VJ$_k$ using specific primers. Each individual lymphoma expressed several related VDJ$_H$ nucleotide sequences, indicating that they had undergone clonal evolution in vivo (Supplementary Fig. 5A–D).

For each lymphoma, a single VDJ$_H$ nucleotide sequence was dominant. Since the lymphomas had a BALB/c background, the dominant sequences were compared with recently reported BALB/c germline V-D-J gene segments[36]. The analysis shows that all four lymphomas expressed somatic mutations in their dominant VDJ$_H$ (Fig. 5A).

Similarly, for each lymphoma, the most frequent VJ$_k$ nucleotide sequence was considered to represent the dominant lymphoma clone (Supplementary Fig. 5). Based on such a likely pairing of dominant VDJ$_H$ and VJ$_k$, there was little sharing of Ig V-related gene segments between the four B lymphomas (Fig. 5B). [The closest IMGT names (based on C57BL/6) for the dominant VDJ$_H$ and VJ$_k$ are given in Supplementary Fig. 6A].

While lymphoma VDJ$_H$ had increased positive charge, increased CDR3 lengths, and increased levels of somatic mutations; these features were largely absent in VJ$_k$ (Fig. 5C–E, and Supplementary Fig. 6B–D).

## Lymphoma BCR H chains preferentially associate with κ chains compared to λ chains

The dual L chain B lymphomas expressed a κ[+] BCR while levels of intracellular λ were increased (Fig. 3L–R). This could suggest that lymphoma BCR H chains preferentially associated with κ chains compared to λ chains. To test this idea, we transfected HEK293 cells with three plasmids encoding, respectively (i) an H chain composed of the dominant lymphoma VDJ$_H$ fused to a human (h)γ3 constant region (ii) an L chain composed of the dominant lymphoma VJ$_k$ fused to hC$_k$ constant region and (iii) a λ L chain composed of VJλ2[315] fused to a hC$_λ$. Presumably, in triple transfectants, an H chain might pair with either lymphoma κ or λ2[315] L chains in the ER, prior to secretion. To reveal any preferential association, levels of respectively Hκ and Hλ Ig in supernatants were analyzed by ELISA. Compared to a control VDJH[315]-hγ3 H chain derived from the BALB/c MOPC315 myeloma (IgA, λ2[315]), the B lymphoma H chains preferred to pair with their endogenous κ compared to the λ2[315] L chain (Fig. 5F).

To further test any preferential association, we performed immunoprecipitations of lysates of cloned NT34.E2 B lymphoma cells and analyzed precipitates by Western blot under non-reducing conditions. Anti-κ precipitated a γ2a H chain while anti-λ did not. The κ was expressed either as a free L chain or as part of a complete IgG2a. In contrast, the λ was expressed only as a free L chain (Fig. 5G, H). The results of the competition and the co-immunoprecipitation

experiments agree with the flow cytometry and confocal microscopy data (Fig. 3L–R) and suggest that in the lymphomas, the H chain prefers to bind κ over λ2[315] and that this happens in the ER where complete Ig is assembled.

## Recombinant Ig corresponding to lymphoma BCRs were autoreactive

To test for autoreactivity of lymphoma BCRs, we made soluble recombinant (r) Ig composed of dominant lymphoma V$_H$ (fused to hγ3) and Vκ (fused to hCκ) for each of the four lymphomas. For three of them, NT8, NT18 and NT24, rIg stained HEp-2 cells with a nuclear staining pattern, while no staining was obtained with the NT7 rIg (Fig. 6A). The three ANA[POS] rIgs bound histones and nucleosomes in ELISAs (Fig. 6B, C) and in line blot assays (Fig. 6D, E) but failed to bind dsDNA (Fig. 6F). Moreover, the ANA[POS] rIg failed to bind a number of other autoantigens (RNP/Sm, Sm, SS-A, Ro52, SS-B, Scl70, PM-Scl100, Jo1, centromere B, PCNA, dsDNA, RIB, M2). The ANA[NEG] NT7 rIg had no reactivity in any of these assays. None of the rIg bound λ2[315] Ig (M315) (Supplementary Fig. 6E).

It was important to ascertain that the rIg corresponded to the lymphoma BCR. An initial experiment demonstrated that the NT34.E2 lymphoma cell line secreted (or shed) the lymphoma BCR in vitro at levels similar to the much-studied A20 B lymphoma cell line (Supplementary Fig. 7A). Based on this result, we injected NT18 (IgG2aκ) lymphoma cells into Ig-deficient Rag 1[−/−] mice. The recipients developed lymphomas, and sera contained IgG2a with ANA activity and reactivity to nucleosomes and histones (Fig. 6G), matching the specificity of rIg derived from NT18 (Fig. 6A). Moreover, proteomic analysis demonstrated that serum from a NT18 Rag 1[−/−] recipient shared V$_H$-derived fragments with NT18 rIg (Supplementary Fig. 7B, C).

## Correlation between (i) specificity of lymphoma BCR expressed as rIg and (ii) specificity of autoantibodies present during the early autoimmune phase

Before overt lymphoma development, during the preceding autoimmune phase, sera of most mice had ANA activity and autoantibodies against histones and nucleosomes (Fig. 1H–P). We therefore tested day 300 sera from the mice that later gave rise to NT7, NT8, NT18 and NT24 lymphomas for the presence of autoantibodies. Autoimmune-phase sera from NT8, NT18 and NT24 mice had positive ANA as well as anti-histone and anti-nucleosome antibodies (Fig. 6A, H, I). Serum from the NT7 mouse contained only low levels of anti-nucleosome and anti-histone antibodies and gave no signal in the HEp-2 assay. These results suggest that B cells present during the autoimmune phase could later give rise to B lymphomas.

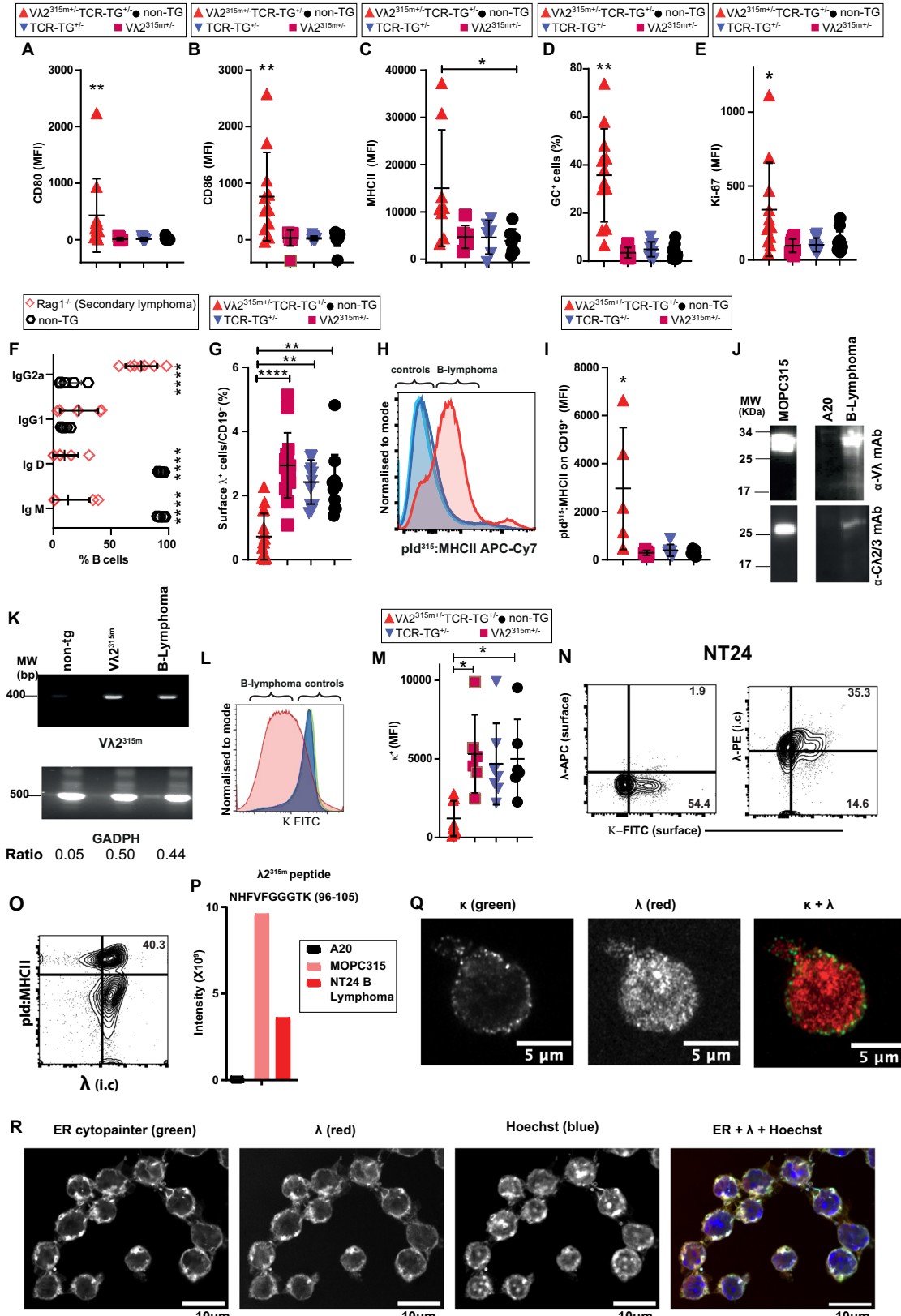

## Mutated B lymphoma $V_H$ sequences were detected in autoimmune phase serum Ig prior to development of overt lymphoma

To obtain more solid evidence that B lymphomas could be derived from B cells during the autoimmune phase, we performed proteomic analysis of lymphoma rIgs and autoimmune-phase serum Igs. Trypsin digestion of the rIgs resulted in fragments that could be readily detected by proteomic analysis. We chose to focus on identifiable $V_H$ fragments that expressed somatic mutations since such fragments should be exceedingly rare among the vastly diversified serum Ig molecules. Such unique $V_H$ fragments are indicated in NT7, 8 and 18 $VDJ_H$ amino acid sequences (Fig. 5A); NT24 lacked such unique

**Fig. 3 | Characterization of B lymphomas. A–D** Expression of surface markers on primary splenic B lymphoma cells in 14–20 months old Vλ2$^{315m+/-}$ TCR-TG$^{+/-}$ mice compared to splenic B cells of age-matched control offspring in cohort one (See Supplementary Fig. 3K for cohort information and Supplementary Fig. 1B for gating of CD19$^+$ B cells). **A** CD80. $p ≤ 0.0016$. Mean (±SD). Vλ2$^{315m+/-}$ TCR-TG$^{+/-}$, $n = 11$; controls, $n = 10$/group. **B** CD86. $p ≤ 0.0149$. Mean (±SD). $n = 11$/group. **C** MHCII. $p = 0.0324$. Mean (±SD). Vλ2$^{315m+/-}$ TCR-TG$^{+/-}$, $n = 10$; controls, $n = 9$/group. **D** GC markers (CD95$^+$GL-7$^+$). $p ≤ 0.005$. Mean (±SD). Vλ2$^{315m+/-}$ TCR-TG$^{+/-}$, $n = 12$; Vλ2$^{315m+/-}$, $n = 8$; TCR-TG$^{+/-}$, $n = 8$; non-TG, $n = 10$. **E** Expression of Ki-67 proliferation marker. $p ≤ 0.0173$. Mean (±SD). Vλ2$^{315m+/-}$ TCR-TG$^{+/-}$, $n = 12$; controls, $n = 11$/group. **F** BCR H chain isotype expression on secondary lymphoma cells established in Rag1$^{-/-}$ mice ($n = 6$) as compared to non-TG B cells ($n = 5$). $p < 0.0001$. Mean (±SD). **G** Frequency of λ$^+$ BCRs in secondary B lymphomas compared to B cells of controls. $p ≤ 0.003$. Mean (±SD). Vλ2$^{315m+/-}$ TCR-TG$^{+/-}$, n = 16; Vλ2$^{315m+/-}$, n = 12; TCR-TG$^{+/-}$, n = 12; non-TG, n = 15. **H, I** Display of pId:MHCII on primary B lymphoma cells, detected by TCRm, as compared to B cells of control mice. $p ≤ 0.0283$. Mean (±SD). $n = 6$/group. **J** Western blots of B lymphoma lysates probed with Vλ and Cλ2/3-specific mAbs.

MOPC315 myeloma (IgA, λ2$^{315}$) and A20 B lymphoma (IgG2a, κ) cell lines served as controls. Eight lymphomas were analyzed with similar results. **K** PCR specific for the mutated λ2$^{315m}$ sequence in secondary B cell lymphoma and B cells of control mice. Shown is one representative example out of 5 lymphomas analyzed. **L, M** Reduced κ expression (κ$^{low}$) in B lymphoma cells compared with B cells of controls. Vλ2$^{315m+/-}$ TCR-TG$^{+/-}$, $n = 8$; controls, $n = 6$/group. **N–R** Characterization of the B lymphoma cell line NT24. One out of three independent experiments is shown. **N** Expression of κ on the cell surface together with either cell surface λ (left) or intracellular λ (right). **O** Surface expression of pId:MHCII and intracellular λ. **P** λ2$^{315m}$-specific peptide levels (given as intensity levels) identified by mass spectrometry. The peptide identified (top) is due to S$^{95}$ → R$^{95}$ conversion in λ2$^{315m}$, generating a trypsin cleavage site. **Q** Confocal microscopy of surface κ and intracellular λ expression. Scale bar: 5 μm. **R** Co-localization of intracellular λ and ER. Scale bar: 10 μm. Statistical comparisons: Unpaired $t$ tests (two-tailed): F; Kruskal–Wallis test (two-sided), Dunn's multiple comparison: A–E, G, I, M. $^*p < 0.05$, $^{**}p < 0.01$ and $^{***}p < 0.0001$. Source data are provided as a Source Data file.

fragments altogether. The selected fragments and their sequences are given in Fig. 6J.

We first demonstrated that the selected fragments were unique for the rIgs from which they were derived, with no cross-reaction to the other rIgs (Fig. 6K). Next, we tested if fragments unique to a rIg could be detected in corresponding sera (purified IgG) from the preceding autoimmune phase. This was generally the case with one exception: fragment number 4 of rIg NT18 was found in NT7 serum (Fig. 6L). One explanation could be that NT7 serum contained an expanded clone of Ig that happened to express the fragment 4 sequence. Taken together, these results indicate that lymphoma progenitor B cells are present already during the autoimmune phase before overt lymphomas developed. It is unclear whether progenitor B cells secreted (or shed) sufficient amounts of antibodies (or BCR) for detection by proteomic analysis. Alternatively, some progenitor B cells could have differentiated into autoantibody-secreting plasma cells.

### Direct support of the hypothesis: A rare B lymphoma expressed α λ2$^{315m+}$ BCR and λ2$^+$ autoantibodies in serum

Summarizing the data presented above, 23/24 B lymphomas expressed a κ$^+$ BCR while λ2$^{315m}$ was only expressed intracellularly. This finding supports the hypothesis of Fig. 1A, but not exactly, since the MHCII-presented Id-peptide in these dual L-chain-expressing B lymphoma cells was derived from an intracellular λ2$^{315m}$ rather than from the BCR (Fig. 6M).

However, we have recently discovered, in a 16-month-old Vλ2$^{315m}$$^{+/-}$ Id-specific TCR-TG$^{+/-}$ mouse, a B lymphoma that expressed a λ$^+$κ$^-$ BCR (Fig. 7A). The primary B lymphoma (NT30) was classified by immunohistochemistry as a CB-DLBCL. It had a phenotype similar to that described above, with increased expression of MHCII, pId:MHCII, CD80/86, and GC (CD95, GL-7) and proliferation (Ki67$^+$) markers (Fig. 7B). Primary NT30 was successfully passaged in Rag1$^{-/-}$ mice. The secondary NT30 was MHCII$^{Hi}$, CD86$^{Hi}$ and displayed pId:MHCII (Supplementary Fig. 8A–C). It expressed an IgG2a BCR (Fig. 7C) and an intracellular κ (Fig. 7D). A Western blot demonstrated that the λ was a λ2/3 which was associated with a γ2a H chain. In contrast, the κ was present only as a free chain (Fig. 7E, and Supplementary Fig. 8D). Secondary NT30 B lymphoma cells induced proliferation of Id-specific CD4$^+$ T cells in vitro (Fig. 7F). These results show that the NT30 lymphoma express a λ2$^+$ BCR, almost certainly due to a Vλ2$^{315m}$→Jλ2 rearrangement.

At the pre-lymphoma stage, the mouse in which NT30 later developed had λ2/3$^+$ antinuclear antibodies (Fig. 7G) with specificity for nucleosomes (Fig. 7H–K) and histones (Supplementary Fig. 8E–H). Proteomic analysis of late lymphoma phase NT30 serum demonstrated that λ-containing Ig expressed Jλ2 but not Jλ1 (Supplementary Fig. 8I). Interestingly, in the autoimmune phase, the NT30 mouse had

not only λ2$^+$ but also κ$^+$ anti-nucleosome/histone autoantibodies. However, by the time of overt lymphoma, λ2 autoantibodies dominated over κ autoantibodies (Fig. 7J–K). These results suggest that malignant transformation could have haphazardly occurred in a rare λ2$^+$ autoreactive B cell within a pool of more numerous κ$^+$ autoreactive B cells.

Taken together, the data suggest that NT30 arose in a κ→λ receptor-revised B cell[37], but where the κ chain replacement by λ2 failed to extinguish autoreactivity. Most likely, the NT30 B lymphoma has a H chain that pairs better with λ2$^{315m}$ L chain than the co-expressed κ chain. These observations directly support the hypothesis of Fig. 1A (Fig. 7L).

### Transfer of Id-specific CD4$^+$ T cells to Vλ2$^{315m+/-}$ mice resulted in B cell lymphoma development

To remove any thymic influence on T cell development, and to test if a more limited number of naïve Id-specific CD4$^+$ T cells could induce lymphomas, we transferred a limited amount of Id-specific CD4$^+$ T cells ($1 × 10^6$) into 2-month-old Vλ2$^{315m+/-}$ mice and followed mice for development of autoimmunity and lymphoma. Non-transferred Vλ2$^{315m+/-}$ mice served as controls (Fig. 7M). 15 months after transfer, Id-specific CD4$^+$ T cells were found in peripheral blood (Fig. 7N). A third of transferred mice (2/6) had at this time point antinuclear antibodies in sera, both of these mice later developed B-cell lymphomas (Fig. 7O, and Supplementary Table 1). Between 16 and 20 months after transfer, two thirds of the transferred mice (4/6) became sick and had to be euthanized while the 4 non-transferred Vλ2$^{315m+/-}$ control mice remained healthy (Fig. 7P, and Supplementary Table 1). Upon euthanasia, 4/6 of transferred mice exhibited macroscopic pathology in lymphoid organs and extra-lymphoid sites (Supplementary Fig. 9A), these four mice were diagnosed with B cell lymphomas by immunohistochemistry (Fig. 7Q, and Supplementary Table 1). In the remaining 2 mice, a lymphoma diagnosis could not be made with certainty (Supplementary Table 1). In sites affected by lymphoma, mice had significant amounts of Id-specific CD4$^+$ Tfh cells and B cells that displayed pId:MHCII and GC markers (Supplementary Fig. 9B, C, and Supplementary Table 1).

## Discussion

We have generated a new model for development of autoimmunity and lymphomas. V gene segment-modified mice express a mutated Id-peptide in a small fraction of their B cells. When crossed with Id-specific TCR transgenic mice, young adult mice developed anti-nuclear autoantibodies. Later, from 12 months of age, mice developed B (and T) cell lymphomas. B lymphoma BCRs had antinuclear activity and were related to preceding serum autoantibodies. These results show that chronic Id-driven T-B collaboration can initiate autoreactive B cell

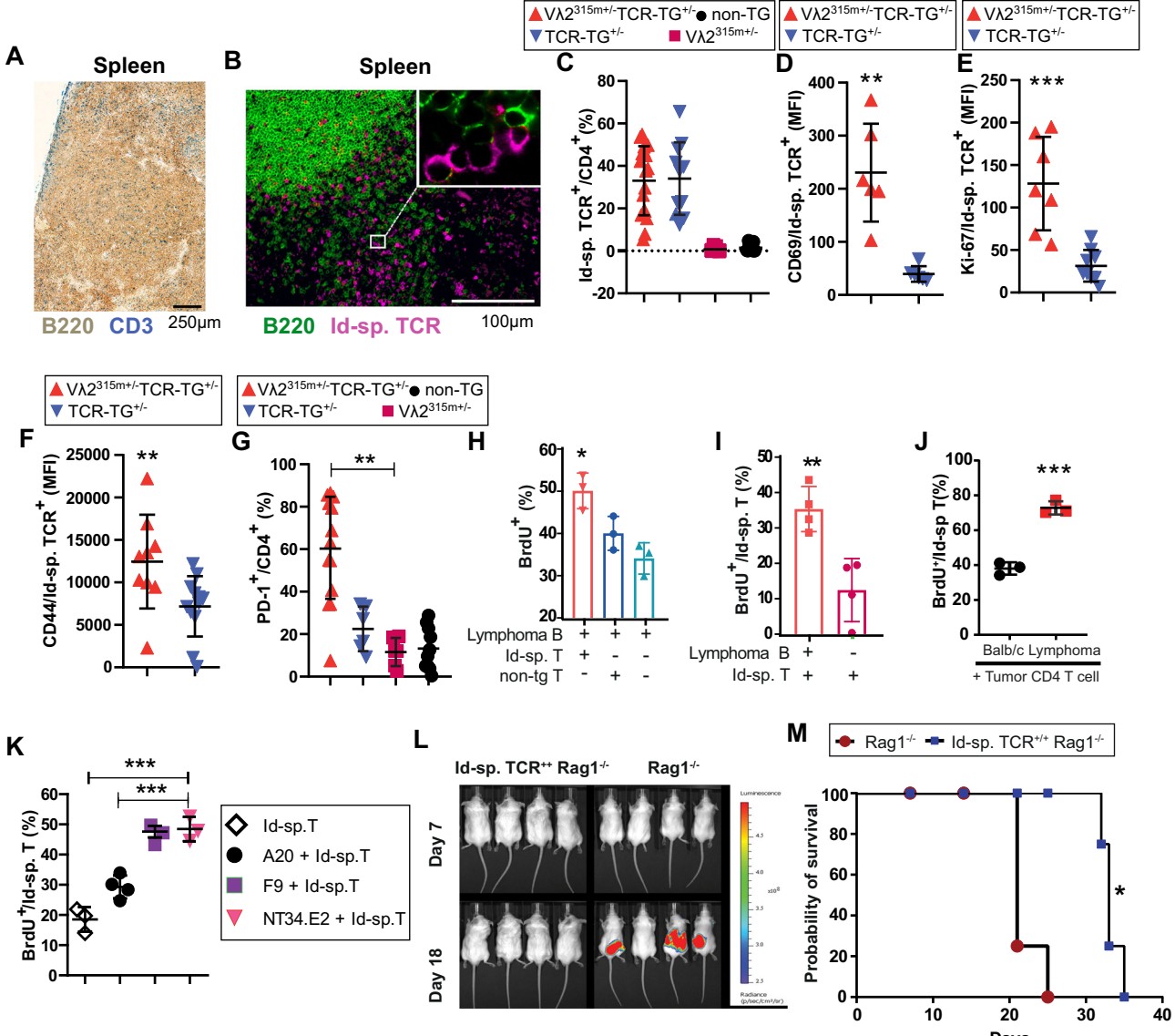

**Fig. 4 | Id-driven T-B collaboration in lymphoma development.**
**A** Immunohistochemistry of a primary splenic lymphoma showing a dominance of B lymphoma cells (B220$^+$) and scattered T cells (CD3$^+$). Scale bar: 250 μm. **B** Frozen section of primary splenic B lymphoma demonstrating proximity between B lymphoma cells (B220$^+$) and Id-specific T cells detected by a TCR clonotype-specific mAb (GB113). Scale bar: 100 μm. **A**, **B** One representative image out of four lymphomas analyzed. **C** Frequency of Id-specific T cells among CD4$^+$ T cells in primary splenic B cell lymphomas compared to spleens of controls. $p > 0.99$. Mean (±SD). Vλ2$^{315m+/-}$ TCR-TG$^{+/-}$, $n = 17$; Vλ2$^{315m+/-}$, $n = 12$; TCR-TG$^{+/-}$, $n = 11$; non-TG, $n = 14$.
**D–G** Phenotype of Id-specific T cells infiltrating B lymphomas compared to Id-specific T cells in spleens of TCR-TG mice (see Supplementary Fig. 1B for gating of Id-specific CD4$^+$ T cells). **D** Expression of CD69. $p = 0.00227$. Mean (±SD). $n = 6/$group. **E** Ki-67. $p = 0.0003$. Mean (±SD). Vλ2$^{315m+/-}$ TCR-TG$^{+/-}$, $n = 7$; TCR-TG$^{+/-}$, $n = 9$. **F** CD44. $p = 0.006$. Mean (±SD). Vλ2$^{315m+/-}$ TCR-TG$^{+/-}$, $n = 9$; TCR-TG$^{+/-}$, $n = 12$. **G** PD-1. $p = 0.0016$. Mean (±SD). Vλ2$^{315m+/-}$ TCR-TG$^{+/-}$, $n = 13$; Vλ2$^{315m+/-}$ and TCR-TG$^{+/-}$, $n = 6$; non-TG, $n = 10$. **H** Irradiated Id-specific T cells from TCR-TG mice induce proliferation (BrdU incorporation) of ex vivo primary B lymphoma cells. Technical replicates ($n = 3$) from one out of two experiments yielding similar results are

shown. $p \le 0.0486$. Mean (±SD). **I** Irradiated ex vivo primary B lymphoma cells stimulate proliferation of Id-specific T cells from TCR-TG mice in vitro. Individual mice from four pooled experiments are shown. $p = 0.0058$. Mean (±SD). **J** Irradiated ex vivo B lymphoma cells stimulate proliferation (BrdU incorporation) of tumor infiltrating CD4$^+$ T cells isolated from the same secondary lymphoma. $n = 3$, technical replicates from a representative experiment out of three. $p = 0.0003$. Mean (±SD). **K** Cloned B lymphoma cells (NT34.E2) induce proliferation (BrdU incorporation) of Id-specific T cells from TCR-TG mice. F9 cells are A20 B lymphoma cells transfected with the λ2$^{315}$ gene[53]. $n = 3$-4/condition, technical replicates. $p < 0.0001$. Mean (±SD). **L** Id-specific TCR-TG Rag1$^{-/-}$ and Rag1$^{-/-}$ mice were injected i.v. with NT34 lymphoma cells labeled with luciferase. In vivo imaging of luciferase activity on indicated days. $n = 4$/group. **M** Survival curves. $^*p = 0.016$. **K**–**M** One representative experiment out of two is shown. Statistical comparisons: Unpaired $t$ tests (two-sided): **I**, **J**; One-way ANOVA (Dunnett's test): **H**, **K**; Mann–Whitney U test (two-sided): **D-F**; Kruskal–Wallis test (two-sided), Dunn's multiple comparison: **C**, **G**; Mantel–Cox test: **M**. $^*p < 0.05$, $^{**}p < 0.01$, $^{***}p < 0.001$ and $^{****}p < 0.0001$. Source data are provided as a Source Data file.

responses that evolve into B cell lymphomas, thus explaining the link between autoimmunity and lymphoma development previously observed in humans[5–7].

Autoimmune B cells and autoreactive B lymphomas[38,39] are likely to receive a stimulatory signal when their BCR recognize a self-antigen (Signal 1). However, for full-blown activation and expansion, these autoreactive B cells need to also receive help from CD4$^+$ T cells (Signal 2). This poses a problem since T cells are tolerant to self-antigens. Solving this conundrum, we here show that autoreactive B cells may receive T cell help from another source: CD4$^+$ T cells that

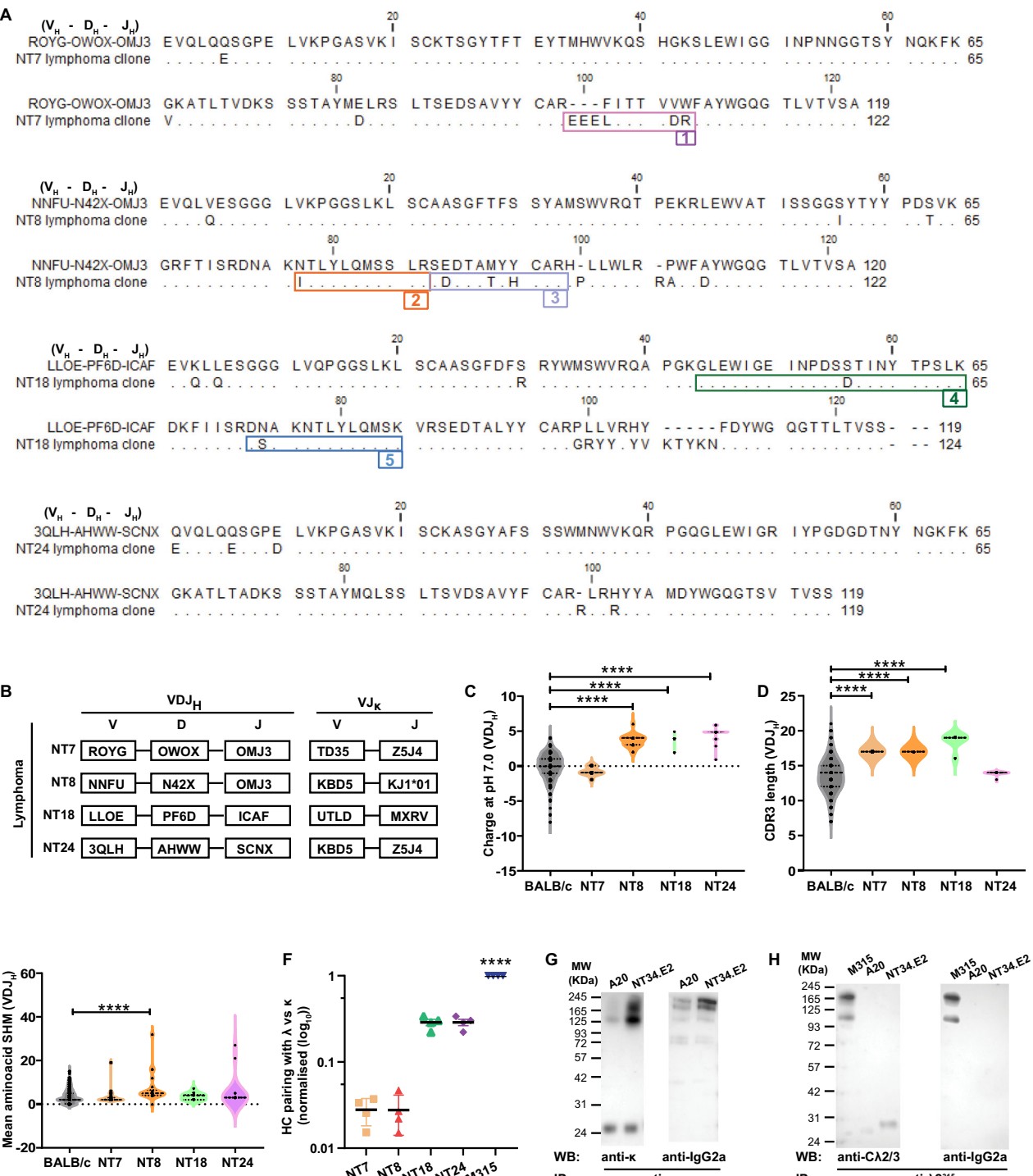

**Fig. 5 | Characterization of lymphoma BCRs. A** Dominant VDJ$_H$ amino acid sequences of secondary B lymphomas NT7, NT8, NT18 and NT24 aligned to the closest BALB/c germline V, D and J sequences[36]. Amino acid replacements due to somatic mutations are indicated. Numbered boxes indicate peptides that were identified by mass spectrometry and that in addition expressed mutations (see Fig. 6J–L). For nucleotide sequences and definition of dominant sequences, see Supplementary Fig. 4. **B** Gene segment utilization of dominant VDJ$_H$ and VJκ of the various lymphomas. Characterization of the dominant VDJ$_H$ of individual lymphomas: net charge at pH 7.0, (**C**) CDR3 length (**D**) and numbers of expressed somatic hypermutations (SHM) (**E**). Number of clones: BALB/c, $n = 463$; NT3, 25; NT8, 22; NT18, 14; NT24, 19. $p < 0.0001$. Violin plots indicate the distribution of individual values. **F** Preferential binding of lymphoma H chain to lymphoma κ or λ2 L chains in an in vivo competition assay where HEK293E cells were co-transfected with three

plasmids encoding respectively (i) lymphoma H chain (ii) corresponding lymphoma k chain and (iii) λ2$^{315}$ chain. For the MOPC15 control (which does not have an endogenous κ), H$^{MOPC315}$ was transfected with a mix of lymphoma k chains and λ2$^{315}$. Levels of Hκ and Hλ expression in supernatants of transfectants were measured. The ratios of Hλ:Hκ were normalized, setting the H$^{315}$λ: H$^{315κ}$=1 [H$^{315}$ from the MOPC315 (IgAλ2$^{315}$) MM cell line presumably pairs well with λ chains]. One representative experiment out of three is shown. $p < 0.0001$. Mean (±SD). **G, H** Western blot analysis. Lysates of indicated lymphoma cells were precipitated with the indicated antibodies, run under non reducing conditions, and blots probed with the indicated antibodies. In **H** an isotype-switched M315 protein (IgG2a, λ2$^{315}$) was used as positive control. One representative experiment out of three is shown. Statistical comparisons: One-way ANOVA (Dunnett's test): **C**–**F**. ****$p < 0.0001$. Source data are provided as a Source Data file.

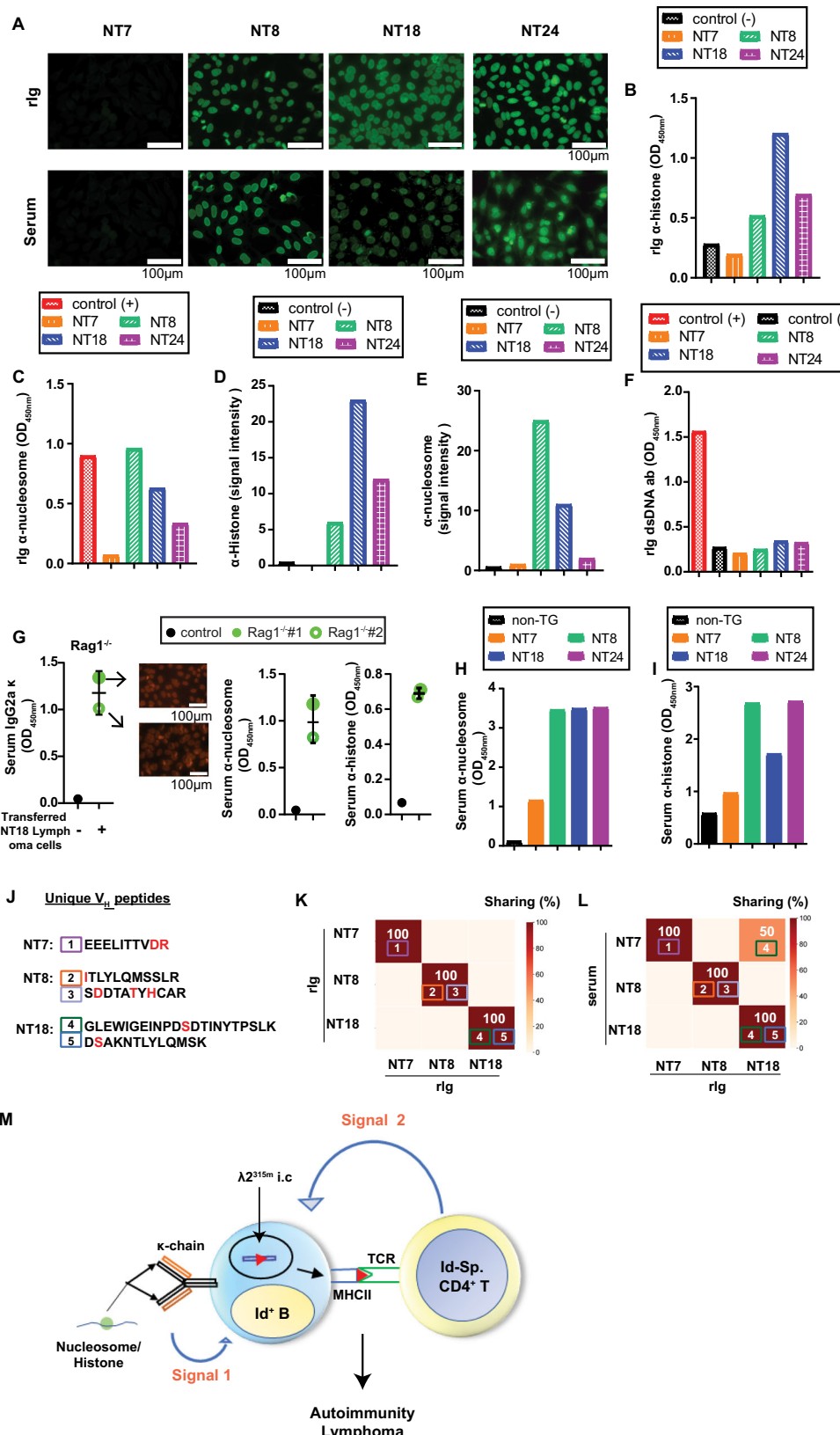

recognize an MHCII-presented neoantigen V-region Id-peptide. The ensuing Id-driven T-B interaction results in a GC reaction since we found that B cells expressed GC markers, had isotype-switched to IgG2a BCR isotype, and expressed mutations in their V regions. Consistent with this, T cells expressed GC-associated Tfh and Tph markers. Following malignant transformation, both the B and T

lymphomas were Bcl6[+], indicating a derivation from GC B cells[40,41] and Tfh[35,42,43], respectively.

The Id-driven T-B collaboration described above is likely to be chronic. This is so because self-antigen (here histone/nucleosome) cannot be removed from the body and are normally found in the extracellular fluid[44]. Also, the Id neoantigen recognized by CD4[+] T cells

**Fig. 6 | Sharing of specificity and VH peptides between recombinant lymphoma BCR and autoantibodies from the preceding autoimmune phase.** The VDJ$_H$ and VJκ from NT7, NT8, NT18 and NT24 lymphomas were expressed together with hγ3 and hCκ, as secreted rIg. **A** Autoreactivity of purified lymphoma rIg in HEp-2 assay was compared with that of autoimmune-phase serum from the corresponding mice. One representative experiment out of four is shown. Reactivity of rIg to histone (**B**), nucleosome (**C**) and dsDNA (**F**) in ELISAs (rIg dilutes 1:4) and in line blots (**D**, **E**). In **B** and **D**–**F**, a mouse anti-CD40 rIg was used as negative control. In **C** and **F**, a positive control included in the kit was used. **G** Sera from two Rag 1$^{-/-}$ mice (filled circle and open circle) carrying NT18 B lymphoma cells, or Rag 1$^{-/-}$ control, were analyzed for IgG2a antibodies staining HEp-2 cells, and binding to nucleosome and histones in ELISA. Mean (±SD). Autoimmune-phase sera (diluted 1:400) from mice giving rise to NT7, NT8, NT18 and NT34 lymphomas were tested for autoantibodies binding to nucleosomes (**H**) and histone (**I**). In **B**–**I** one representative experiment out of two is shown. **J** Numbering and sequences of mutated V$_H$ fragments derived from NT7, NT8 and NT18. **K** The selected V$_H$ fragments (see **J**) are unique to the rIg they are derived from. **L** Detection of selected V$_H$ fragments (see **J**) in purified IgG from autoimmune-phase sera. **M** Summary of data from Figs. 1–5. κ$^+$ BCR is autoreactive while the λ2$^{315m}$, yielding the MHCII-presented Id-peptide, is only expressed intracellularly. Illustration was made with Microsoft PowerPoint. Source data are provided as a Source Data file.

cannot easily be extinguished except if all cells of a B clone are killed, or additional V region mutations abolish expression of the Id neoantigen. The latter is unlikely to occur in all cells of a B cell clone. In contrast, in conventional T-B cell collaboration, the extrinsic antigen involved usually disappears with time (except in chronic infections) causing B and T cell stimulation to subside.

The chronic proliferation of autoreactive B cells helped by Id-specific CD4$^+$ T cells could facilitate acquisition of oncogenetic events and finally malignant transformation. The mouse B lymphomas generated herein correspond to major types of human B lymphomas including DLBCL and FBL[33]. Development of different types of B lymphomas could be related to the type of secondary genetic changes that a chronically proliferating B cell happens to undergo[3,45–47]. Since a large number of different B lymphomas were easily generated, the present model might become a valuable tool for studying genetic events involved in malignant transformation.

The model can explain the link between autoimmunity and lymphoma development previously observed in humans[5–7]. Thus, BCR rIg and serum autoantibodies shared specificity for histone/nucleosome. Moreover, proteomic analysis demonstrated that mutated V region fragments were shared between the lymphoma BCR and preceding autoimmune phase antibodies in a single mice. Finally, like in humans, female mice were more prone to develop autoimmunity[48] with more frequent progress to B lymphomas. Thus, the present mouse model reflects features of human autoimmunity and B lymphoma development.

The present findings may offer a mechanistic explanation for why Id-specific CD4$^+$ T cells have been found in several human autoimmune diseases such as SLE[16], rheumatoid arthritis[17], and multiple sclerosis[18]. An important question is whether the tumor-enhancing effect of Id-specific CD4$^+$ T cells could also operate in humans. Khodadoust et al. found that MALT B lymphomas[20] as well as DLBCL, FBL, and B-CLL[21] efficiently presented mutated Id-peptide neoantigens on their MHC class II molecules. The authors presented circumstantial evidence suggesting that pId:MHCII complexes on lymphoma cells could be targets for CD4$^+$ cytotoxic T cells[20]. The present findings favor an alternative explanation: malignant B cells could be selected for expression of high amount of pId:MHCII simply because they would then be more apt to solicit help from Id-specific CD4$^+$ T cells.

To our surprise, the great majority of lymphoma BCR did not carry the λ2 Id neoantigen but rather a κ L chain. Instead, the λ2 was expressed intracellularly in the ER as a free L chain. Dual L chain B cells have previously been associated with autoimmunity in humans[49] and mice[50–52]. Dual L chain expression may be explained by preferential H-L chain association during development of a B cell in the bone marrow[52]. However, and importantly, we found that 1 out of 24 B lymphomas had a λ2$^+$ BCR. Moreover, the mouse in which this lymphoma arose had serum λ2$^+$ autoantibodies. This observation directly supports the hypothesis that Id-driven T-B collaboration can cause autoimmunity and B lymphoma development (see Fig. 1A). Most likely, the H chain of this rare B lymphoma pairs well with λ L chains. The present results are consistent with previous results indicating that two pathways exist for B cell processing of Ig V- regions and presentation on MHCII: (i)

nascent Ig chain in ER/pre Golgi[53], and (ii) ligation of BCR subsequent to ligation by antigen[26].

It is striking that 23/24 (96%) of B lymphomas probably have undergone a λ2$^{315m}$→κ revision while 1/24 (4%) is likely to have undergone the reverse revision, κ→ λ2$^{315m}$. These frequencies correspond roughly to k/λ ratios in mouse serum Ig. The H chain that happens to be expressed early in a B cell's development could influence this result by preferentially pairing with either κ or λ L chains. It is also striking that we did not find any B lymphomas that only expressed one L-chain, i.e., λ2$^{315m}$. We speculate that this could be related to the reinstated Rag expression in dual L chain B cells that somehow could be lymphomagenic.

Id-driven T-B collaboration is difficult to study; hence, reductionistic models are needed. Herein, we used a V gene segment-modified mouse that permits physiological expression of a λ2 chain with a mutated CDR3 sequence at a normal developmental time point and only in a small number of B cells (0.5%)[26]. This low number of B cells induced a partial deletion of Id-specific CD4$^+$ T cells in the thymus and periphery, and development of FoxP3$^+$ Id-specific CD4$^+$ T cells in the spleen. Nevertheless, Id-specific CD4$^+$ T cells were fully responsive to Id peptide in vitro. By extension, in normal mice, a clone of B cells expressing somatic mutations is unlikely to induce full-blown T cell tolerance. Excluding a necessity of thymus-dependent modification of T cell phenotype for lymphomagenesis, B lymphomas could be induced by i.v. transfer of Id-specific CD4$^+$ T cells into V gene segment-modified mice. Moreover, a single transfer of a relatively modest number (1×10$^6$) of Id-specific CD4$^+$ T cells was sufficient for B lymphoma development, thus making the model more physiological also on the T cell side.

The growth of B lymphoma cells was retarded when injected into Id-specific TCR-transgenic mice. This demonstrates that CD4$^+$ T cells, having an identical TCR, could elicit B lymphomas in one experimental situation, while in another situation, could curb lymphoma growth. These two opposing effects could be due to different CD4$^+$ T cell subtypes operating in the two experimental settings. In lymphoma rejection, a Th1/M1 macrophage-mediated mechanism operates[54–58]. By contrast, in the present lymphomagenesis study, Tfh cells secreting IL-21 appear to be important. These could be Tfh1 cells[31] since all B lymphomas expressed an IgG2a BCR, and since lymphoma mice had increased serum levels of IFN-γ and TNF. In later stages, lymphomas were infiltrated by PD-1$^+$ CXCR5$^-$ CD4$^+$ T cells; these could either be peripheral helper T (Tph) cells[29,30] or exhausted T cells[59]. In humans, dysregulated Tfh activity has been linked to follicular lymphoma (FL) and diffuse large B cell lymphoma (DLBCL)[60,61] while Tfh1 has been associated with marginal zone lymphoma (MZL)[62].

Development of T cell lymphomas was also frequent in the present model, but we did not study these in detail. T cell lymphomas expressed Bcl6, which suggests that they could originate from Id-specific Tfh cells that proliferated extensively in response to autoreactive Id$^+$ B cells. Even though microscopic analysis of sections could indicate coexistence of both B and T lymphomas in a single mouse, we failed to obtain both types of lymphomas after transfer of a primary lymphoma to Rag1$^{-/-}$ mice.

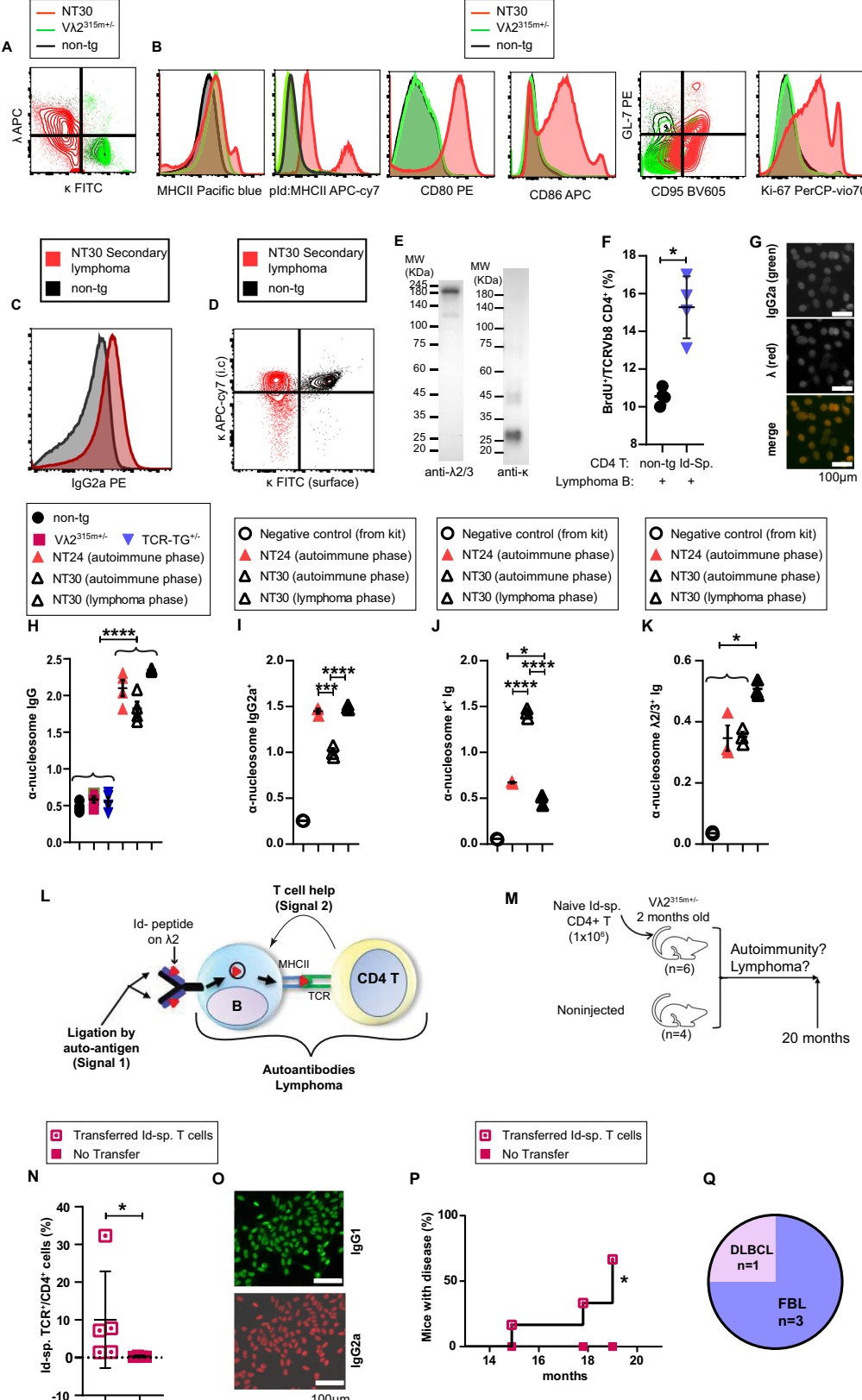

Id-driven T-B collaboration involves (i) an autoreactive B cell and (ii) a CD4⁺ T cell specific for Id-peptide neoantigen presented on MHCII. Such Id-driven T-B collaboration depends upon several chance events like particular rearrangements of Ig genes in B cells and TCR genes in T cells, somatic mutations of rearranged V(D)J gene segments in B cells, and encounter of rare B and T cells in lymphoid organs. Finally, Id-driven T-B collaboration is likely to be suppressed in normal mice and humans. Failure of suppression could result in extensive proliferation of both B and T cells, autoimmunity and finally development of B and T cell lymphomas.

**Fig. 7 | A B lymphoma that expresses a λ2³¹⁵ᵐ BCR. A–B** Characterization of the primary NT30 B lymphoma that arose in a 16-month-old Vλ2³¹⁵ᵐ⁺/⁻Id-specific TCR-TG⁺/⁻ mouse. **A** Spleen CD19⁺ B lymphoma cells display cell surface λ but not κ. **B** Expression of MHCII, pId:MHCII, CD80, CD86, GC markers (CD95⁺GL7⁺) and Ki-67 in B lymphoma cells. **C–F** Characterization of secondary NT30 B lymphoma passaged in Rag1⁻/⁻ mice. Spleen CD19⁺ B lymphoma cells display surface IgG2a (**C**) while κ is expressed i.c. (**D**). **E** Western blot under non-reducing conditions demonstrating a free κ while λ associated with a γ2a H chain (see Supplementary Fig. 8D). One of two independent experiments is shown. **F** Irradiated primary spleen CD19⁺ NT30 cells were cultured with CD4⁺ T cells from either TCR-TG or BALB/c mice. Proliferation (BrdU incorporation) of TCRVβ8⁺ CD4⁺ T cells is shown. p = 0.0286. Mean (±SD). n = 4, technical replicates, one out of two independent experiments. **G** λ⁺ANA in serum of the pre-lymphoma (12-month-old) primary NT30 mouse. Scale bar:100 μm. Nucleosome-specific autoantibodies in sera of the NT30 mouse, obtained either during the pre-lymphoma autoimmune phase (12 months) or the overt lymphoma phase (16 months). Shown are IgG (**H**), IgG2a (**I**), κ (**J**), and λ2/3 (**K**). The control NT24 autoimmune phase serum contains predominantly κ⁺ α-

nucleosome/histone autoantibodies typical of κ⁺B lymphomas. n = 4/group (**H**) or n = 3/group (**I–K**), technical replicates. One out of two independent experiments is shown. p-values: **H**, p < 0.0001; **I**, p = 0.0001 (***) and p < 0.0001 (****); **J**, p = 0.0114 (*), ≤ 0.008 (***) and p < 0.0001 (****), **K**: p ≤ 0.0253 (*). Mean (±SD). **L** Schematic drawing showing that Id-driven T-B collaboration, where pId is derived from λ2⁺ BCR, induces autoimmunity and B lymphoma. The illustration was made with Microsoft PowerPoint. **M–Q** Injection i.v. of naïve Id-specific T cells into Vλ2³¹⁵ᵐ⁺/⁻ mice elicits autoantibodies followed by B lymphoma development. **M** Outline of experiment. Two-month-old Vλ2³¹⁵ᵐ⁺/⁻ mice were injected with CD4⁺ spleen T cells from TCR-TG mice. Control Vλ2³¹⁵ᵐ⁺/⁻ mice did not receive T cells. **N** Expansion of Id-specific CD4⁺ T cells in the peripheral blood 13 months after transfer. p = 0.0159. Mean (±SD). **O** Two out of 6T cell-transferred mice developed ANA. Scale bar: 100 μm. One out of two independent experiments is shown. **P, Q** Four out of 6T cell-transferred mice developed the indicated type of B lymphoma. **P:** p = 0.022. Statistics: **F** and **N**: Mann–Whitney U test (two-tailed); **H–K**: One way ANOVA test (Tukey's multiple comparison); **P**: Mantel–Cox test. *p < 0.05, ***p < 0.001, ****p < 0.0001. Source data are provided as a Source Data file.

## Methods

### Mice, typing, and breeding
The Vλ2 gene-modified Vλ2³¹⁵ᵐ mice[26] and the Id-specific TCR-transgenic (TG) mice were maintained in a heterozygous state on a BALB/c background. For experiments, Vλ2³¹⁵ᵐ⁺/⁻ and Id-specific TCR-TG⁺/⁻ were crossed, thus obtaining four types of offspring: Vλ2³¹⁵ᵐ⁺/⁻ Id-specific TCR-TG⁺/⁻, Vλ2³¹⁵ᵐ⁺/⁻, Id-specific TCR-TG⁺/⁻ and non-transgenic mice. The Vλ2³¹⁵ᵐ V gene-modification[26] was typed by PCR while the TCR-TG mice were typed by flow cytometry using the TCR clonotype-specific GB113 mAb[63]. The breeding was set up as (Vλ2³¹⁵ᵐ⁺/⁻ ♀ x Id-specific TCR-TG⁺/⁻ ♂) for the first two cohorts. For the third cohort (Id-specific TCR-TG⁺/⁻ ♀ x Vλ2³¹⁵ᵐ⁺/⁻ ♂) were used. Rag1⁻/⁻ (C.129S7(B6)-Rag1tm1Mom/J, The Jackson Laboratory, (stock #002216)) and Id-specific TCR-TG⁺/⁺Rag1⁻/⁻ mice were maintained on a BALB/c background and were used in transfer and tumor challenge experiments. BALB/c mice were purchased from Janvier labs (strain identifier: BALB/cAnNRj). Animals were housed in specific pathogen-free (SPF) unit at 20–23 °C and 45–65% humidity with a 12-h dark/light cycle. Both the experimental and control animals were co-housed throughout the study. The mouse experiments were approved by the Norwegian Food Safety Authority (ID 12387, 27752). Experimental procedures followed institutional guidelines of the Department of Comparative Medicine, Oslo University Hospital.

### Tissue culture medium
Cells were cultured in RPMI 1640+ GlutaMAX (Gibco) supplemented with 10% heat-inactivated fetal calf serum (Biochrom and Sigma-Aldrich) to which were added 0.1mM NEAA (Invitrogen), 1 mM sodiumpyruvate (Invitrogen), 0.5 mM 1-thioglycerol (Sigma-Aldrich) and 20 μg/ml Gensumycin (Sanofi-Aventis Norway AS).

### Transfer of Id-specific CD4⁺ T cells into Vλ2³¹⁵ᵐ⁺/⁻ mice
Id-specific CD4⁺ T cells were isolated from young TCR-TG⁺/⁻ by negative selection using CD4⁺ T cell Isolation Kit, mouse (Miltenyi Biotec, #130-104-454). Among the purified CD4⁺ T cells, ~50% were Id-specific T cells. Two-month-old Vλ2³¹⁵ᵐ⁺/⁻ mice were injected with 2 × 10⁶ CD4⁺ T cells amounting to ~1 × 10⁶ Id-specific T cell. Age-matched Vλ2³¹⁵ᵐ⁺/⁻ control mice did not receive any T cells.

### Follow-up of mice, dissection and tissue processing
Weekly follow-up of mice consisted of weighing, physical examination including abdominal palpation, and assessment of fur condition and motility. The humane endpoint was defined based on several criteria such as mobility, posture, weight loss and increase in abdominal size. Upon reaching the endpoint, mice were euthanized by intraperitoneal administration of ZRF (0.1 ml/10 g body weight; Zolazepam 3.29 mg, Tiletamine 3.29 mg, Xylazine 0.45 mg, Fentanyl 2.6 mg) followed by a

secondary methods of euthanasia, such as exsanguination or cervical dislocation, dissected and evaluated for enlargement of the spleen, the thymus and the lymph nodes (LNs). Spleens were collected into GentleMACS C tubes (Miltenyi Biotec) with RPMI tissue culture medium supplemented with 1 mg/ml of collagenase type 4 (Worthington, #LS004188) and 0.3 mg/ml DNaseI (Sigma-Aldrich, #D5025). The spleens were dissociated with a GentleMACS Dissociator (Miltenyi Biotec) and incubated with collagenase/DNase for 10 min at 37 °C. Single cell suspensions were washed in complete medium and incubated with Tris-buffered ammonium chloride (ACT) for 7 min on ice to lyse erythrocytes. Following ACT treatment, the cells were washed again in complete medium and viable cells were counted using a cell counter (Countess, Invitrogen).

LNs and thymi were collected in tubes containing tissue culture medium and pressed with a syringe piston through a cell strainer. The cells were washed in complete medium and viable cells were counted.

Tissue pieces were fixed in 5% neutral-buffered formalin or frozen in isopentane chilled on dry ice.

### Histopathology and immunohistochemistry of lymphomas
Histopathological examination was done on formalin-fixed, paraffin-embedded sections (FFPE) stained with hematoxylin and eosin. Guidelines for evaluation were as published[33]. Frozen sections were used for co-staining of T and B cells using anti-CD3 mAb (clone CD3-12, Bio-Rad, #MCA1477T, 1:100) and anti-B220 mAb (clone RA3-6B2, BD Biosciences, #553090, 1:200). For staining of Id-specific TCR⁺ cells, frozen lymphomatous spleen sections were cut at 5 μm, air-dried, fixed in acetone, washed, and blocked with 30% normal rat serum and 30 μg/ml of FcRγ-blocking mAb (HB197, ATCC) in PBS/BSA (30 min at 25 °C). Sections were then incubated with anti-B220-FITC mAb (RA3-6B2, TONBO Biosciences, #SKU 35-0452-U025, 1:200), anti-Bcl-6 pAb (MBS9215821, MyBioSource, #MBS9215821, 1:25) and GB113-PE (clonotype-specific for the Id-sp TCR[63], Diatec Monoclonals, 1:100) for 1 h at 25 °C. Secondary detection reagents were rabbit AF488-coupled anti-fluorescein (#A-11090, Thermo Fisher Scientific, 1:200) and goat anti-PE-Texas Red (#ab34734, Abcam, 1:1000), applied for 45 min at 25 °C. FFPE sections were used to stain Pax-5, IBA-1 and CD3 using the following antibodies: anti-Pax-5 pAb (#LS-C88806-01, LSBio, 1:100), anti-IBA-1 pAb (#hs-234 013, Synaptic Systems GmbH, 1:200) and anti-CD3 pAb (#sc-1127, Santa Cruz Biotechnology, 1:100). Images were acquired using a DS-Ri1 camera and captured using NIS-Elements F4.30.01 software.

### Staining of cells and flow cytometry
Single cell suspensions were stained by a number of different fluorochrome-labeled mAbs: Anti-BrdU APC from kit (#552598, BD Biosciences, 1:100), Anti-BrdU APC (clone Bu20a, Biolegend, catalogue

#339808, 1:20) anti-CD19 PE-Cy7 (6D5, Biolegend, #115520, 2.5 μg/ml), anti-κ-APC-Cy7 (RMK-45, Biolegend, # 409504, 2.5 μg/ml), anti-κ-FITC (187.1, Thermo Fisher Scientific, #OB1170-02, 8 μg/ml), anti-λ-APC (RML-42, Biolegend, #407306, 4 μg/ml), anti-CD80-APC (16-10A1, Biolegend, #104714, 8 μg/ml), anti-CD86-PE (GL1, Southern Biotech, # 735-09 L, 4 μg/ml), anti-mouse I-A/I-E-FITC (2G9, BD Biosciences, #553623, 8 μg/ml), anti-Ki-67-PerCP-Vio700 (REA 183, Miltenyi Biotec, #130-120-418, 2 μl/test), anti-CD95-FITC (Jo2, BD Biosciences, #554257, 8 μg/ml), anti-CD95-BV605 (SA367H8, Biolegend, #152612, 2.5 μg/ml), anti-GL7-FITC (GL7, Biolegend, #144604, 8 μg/ml), TCRm-biotin (anti-pId:MHCII scFv TCR mimetic,)[26] (homemade, 1.5ug/ml), anti-CD4-APC (GK1.5, Southern Biotech, #1540-11, 4 μg/ml), anti-CD4-AF405 (RM4-5, Thermo Fischer Scientific, #MCD0426, 4 μg/ml), anti-CD69-PerCP/Cy5.5 (H1.2F3, Biolegend, #104520, 3 μg/ml), anti-mouse TCRvβ 8.1-3-FITC (F23.1, Santa Cruz Biotechnology, #sc-33648, 1:5), anti-Id-specific TCR mAb GB113 (homemade, 8 μg/ml), anti-CD93-PE (AA4.1, Biolegend, #136504, 3 μg/ml), anti-CD21/CD35-FITC (7G6, BD Biosciences, #553818, 8 μg/ml), anti-FoxP3-APC (FJK-16s, Thermo Fisher Scientific, #17-5773-82, 8 μg/ml) anti-IgM-APC (RMM-1, Biolegend, #406509, 8 μg/ml), anti-CD23-PE-Cy7 (B3B4, Thermo Fisher Scientific, #25-0232-82, 2 μg/ml), anti-CD44-PE-Cy7 (IM7, Biolegend, #103030, 3 μg/ml), anti-PD-1-APC-Cy7 (29 F.1A12, Biolegend, #135224, 1.5 μg/ml), anti-CXCR5-BV421 (L138D7, Biolegend, #145512, 6 μg/ml), goat anti mouse-IgG1-AF488 (polyclonal, Thermo Fisher Scientific, #A-21121, 2 μg/ml), goat anti mouse-IgG2a (polyclonal, Thermo Fischer Scientific, #A-21134, 2 μg/ml), anti-IgG2b (G15-337, BD Biosciences, #553884, 2 μg/ml), anti-CD45R-PerCP/Cy5.5 (RA3-6B2, BD Biosciences, #552771, 4 μg/ml), anti-λ2/3-biotin (2B6, homemade, 6 μg/ml)[64], anti-κ-biotin (187.1, homemade, 3 μg/ml). The following secondary reagents were used in combination with biotinylated antibodies: streptavidin-APC/Cy7 (#405208, Biolegend, 1:300), streptavidin-PerCP (#554064, BD Biosciences, 1:500) and streptavidin-PE (#554061, BD Biosciences, 1:1000). Labeled mAbs were diluted in PBS (#D8537, Sigma-Aldrich) containing 0.5% BSA (#805095, Bio-Rad) with 30% normal rat serum (#GTX73226, GeneTex) and FcRγ blocker mAb. Stainings were done as described in ref. 26. For intracellular stainings, BD Cytofix/Cytoperm (#554714, BD Biosciences) kit was used. For FoxP3 staining, BD Pharmingen™ Transcription Factor Buffer Set (#562574) was used. In vivo proliferation was assessed using the Ki-67 marker (mAb clone REA183, #130-100-339, Miltenyi Biotec). Ki-67 staining was done as described in ref. 65. Stained cells were run on a BD LSR II instrument and Attune NxT (from Thermo Fisher Scientific). Compensation was performed using OneComp eBeads (Thermo Fisher Scientific, #01-1111-42) stained individually with each antibody used in the panel. The data were analyzed using FlowJo_V10.6.1 (TreeStar, Ashland, OR) software.

## Transfer of lymphoma cells to Rag1$^{-/-}$ mice
Rag1$^{-/-}$ mice were injected *i.v.* with $1 \times 10^6$ cells isolated from the spleens of mice with suspected primary lymphomas. The cancerous state of suspected primary lymphomas was always confirmed by establishing secondary lymphomas in Rag1$^{-/-}$ mice. Upon reaching the humane endpoint, the Rag1$^{-/-}$ recipients were dissected and lymphomatous tissues collected for IHC and for FACS staining.

## In vitro BrdU proliferation assays
To study the proliferative T cell responses to synthetic peptides, Splenocytes from BALB/c were irradiated with 20 Gy. CD4$^+$ T cells were isolated from 4 weeks old Vλ2$^{315m+/-}$ Id-specific TCR-TG$^{+/-}$ and Id-specific TCR-TG$^{+/-}$ mice by negative selection using immunomagnetic depletion Dynabeads Mouse CD43 (Untouched B cell Kit, #11422D, Thermo Fisher Scientific) and Dynabeads Untouched™ Mouse CD4 Cells Kit, #11415D, Thermo Fisher Scientific). $1 \times 10^6$ splenocytes and $1 \times 10^5$ Id-specific T cells were added to a total volume of 1 ml complete medium in 48 well plates and stimulated with 10 μg/ml synthetic λ2 peptides [FAALWFRNHFVFGGGTK (λ2$^{315}$) or FAALWYSTHYVFGGGTK

(germline λ2), Genscript] for 48 h. BrdU was then added to the culture to obtain a final concentration of 1 mM. After 16 h cells were harvested, stained and analyzed by FACS.

To study the proliferative responses of ex vivo B and T cells obtained from tumor tissues, B cells and CD4 T cells were sorted from lymphomatous spleens or BALB/c spleens by negative selection using immunomagnetic depletion. B and T cells ($5 \times 10^5$/of each in 1 ml, using 48 well plates) were co-cultured in complete culture medium for 72 h; the last 16 h in the presence of BrdU (1 mM). Incorporation of BrdU was detected using a BrdU Flow Kit (#552598, BD Biosciences). Briefly, 16 h after pulsing with BrdU, the cells were harvested, washed and surface stained. Then the cells were fixed with BD cytofix/cytoperm for 15 min on ice. The cells were washed with BD perm/wash and incubated with BD Cytoperm Permeabilization Buffer for 10 min on ice. The cells were washed again and refixed with BD cytofix/cytoperm. The cells were treated with DNase I for 1 h at 37 °C to expose BrdU. The cells were washed and stained with anti-BrdU APC diluted in BD perm/wash and washed. The cells were then analyzed by flow cytometry.

## In vivo BrdU incorporation experiments
Age matched 14 months old Vλ2$^{315m+/-}$, Vλ2$^{315m+/-}$ Id-specific TCR$^{+/-}$ and Id-specific TCR$^{+/-}$ mice were injected with 1 mg of BrdU (#B5002, Sigma-Aldrich) i.p. on day 1. Thereafter, mice received BrdU (0.6 mg/ml) in the drinking water for 7 days before euthanasia and analysis. Single cells from spleen and LN were surface stained and fixed with BD cytofix/cytoperm for 15 min on ice. The cells were then treated with BD Cytoperm™ Permeabilization Buffer Plus for 5 min (#561651, BD Biosciences) followed by incubation with DNase I at 37 °C for 1 h. Finally, the cells were stained with anti-BrdU-APC mAb (clone Bu20a, Biolegend) at 25 °C for 30 min and analyzed by flow cytometry.

## Lymphoma cell lines and clones
Single cells from spleens of Rag1$^{-/-}$ mice transferred with primary B lymphoma cells were cultured in vitro in RPMI supplemented with an increased amount of FCS (20%). Generally, the cells grew only for a few days before dying. The cells from mouse #24 (NT24), however, grew for a few passages and were used for confocal microscopy and flow cytometry. Only cells from lymphoma NT34 proliferated continuously to give rise to the cell line NT34. The cells which expressed the highest amount of λ L chains intracellularly were selected by limiting dilution to obtain a cell line called NT34.E2. The FCS level could be reduced to 10% with maintenance of cell growth. NT34 cells were transiently transfected with lentiviral transduction, using the bicistronic expression vector pHIV-Luc-ZsGreen, encoding firefly luciferase and the green fluorescent protein ZsGreen. Then the GFP$^+$ cells were sorted on a BD FACS Aria. Sorted cells were called NT34.Effluc-GFP and were used in tumor challenge experiments.

The T cell lymphomas were easier to grow compared to the B lymphomas. The spleen cells of secondary lymphomas in Rag1$^{-/-}$ mice were cultured with 100 IU/ml IL-2. After few passages, the T cells started to grow without IL-2 supplement.

## Tumour challenge and IVIS
$0.5 \times 10^6$ NT34.Effluc-GFP cells were injected i.p. into Rag1$^{-/-}$ and Id-specific TCR-TG$^{+/+}$ Rag1$^{-/-}$. The mice were injected i.p. with D-Luciferin (150 mg/kg body weight, Sigma-Aldrich) and anesthetized with isoflurane. 10 min later, images reflecting tumor load were acquired on days 7, 14, 18, 21, 28 and 35 using an IVIS Spectrum imaging system (Caliper Life Sciences). Data were analyzed using the LivingImage software (Caliper Life Sciences).

## Western blot
Cells were lysed with NP-40 lysis buffer containing a protease inhibitor cocktail (P8340, Sigma-Aldrich). Samples were denatured with sample buffer containing 2-Mercaptoethanol (#M3148, Sigma-Aldrich) and

Dithiothreitol (#D0632, Sigma-Aldrich) at 95 °C for 5 min, run on a 4-12% Tris-bis gel (NW04125BOX, Novex, Carlsbad, CA, USA), and transferred to a PVDF membrane (#IB24001, Invitrogen). The membrane was blocked with 3% milk powder (#A0830, AppliChem, Darmstadt, Germany) in PBS containing 0.05% Tween. The λ Ig L chain was detected with following antibodies: Cλ2/Cλ3 L chain cross-reactive rat mAb 2B6[64] (1 μg/ml) and Vλ1/Vλ2 cross-reactive rat mAb 9A8[64] (1 μg/ml). Primary antibodies were detected with goat anti-rat IgG Horseradish Peroxidase (HRP) (#31470, Thermo Fisher Scientific, 1:5000) and developed with ChemiGlow West Chemiluminescence Substrate Kit (#60-12596-00, ProteinSimple). In co-precipitation experiments, the cell lines were lysed with NP-40 lysis buffer containing 0.1% CHAPS (#C9426, Sigma-Aldrich). 30 μl of protein G beads were coated with 1 μg of anti-λ2[315] R/A (Rabbit pAb)[53] or anti-κ mAb (H139-52.1, Sothern Biotech, #1180-01, 1:2000) and blocked with rat serum. Then λ and κ Ig were precipitated from lysates and analyzed by Western blots under non-reducing condition using the biotin labeled mAb: anti-Cλ2/cλ3 (2B6, 1 μg/ml), anti-κ (187.1, 1 μg/ml), anti-IgG2a[a] (8.3, BD Biosciences, #553502, 1:1000), followed by streptavidin-HRP conjugate (#7100-05, Southern Biotech, 1:5000). Blots were captured using G:box Chemi XX6 device (SYNGENE).

## Cytokine multiplex and IL-21 ELISA

23 different cytokines in sera were analyzed using Bio-Plex Pro Mouse Cytokine 23-plex Assay (M60009RDPD, Bio-Rad). The plates were read on Bio-Plex MAGPIX. IL-21 ELISA was performed using IL-21 Mouse Uncoated ELISA Kit (8-8210-88, Invitrogen).

## Mass spectrometry

For immunoprecipitation, 30 μl of Dynabeads Protein G for Immunoprecipitation (10003D, Thermo Fisher Scientific) were washed and coated with 1 μg of anti-Vλ1/Vλ2 mAb (9A8) at 25 °C. To precipitate serum IgG, uncoated protein G beads were used. Precipitated proteins on beads were dissolved in 50 mM ammonium bicarbonate, reduced, alkylated and digested with 1 μg AspN (Promega) for 1 h and then 1 μg trypsin (Promega) at 37 °C overnight. Digested peptides were acidified and desalted by homemade C18 stage tips. LC-MS/MS analysis was carried out using a nanoElute nanoflow ultrahigh-pressure LC system (Bruker Daltonics, Bremen, Germany) coupled to the timsTOF fleX mass spectrometer (Bruker Daltonics), using a CaptiveSpray nanoelectrospray ion source (Bruker Daltonics). Raw files from LC-MS/MS analyzes were submitted to MaxQuant 2.4.7.0 software for protein identification and label-free quantification. A custom database having lymphoma BCR sequences was used.

## Super-resolution immunofluorescence microscopy

Coverslips were sterilized with 95% EtOH, flamed and placed in 6-well plates. The coverslips were treated with poly-L-lysine for 10 min at 37 °C. The coverslips were then washed. Lymphoma B cells were cultured in RPMI w/ 20% FCS media for 2 days. Before staining, the wells were washed with PBS, and the cells were fixed with 2% formaldehyde (F8775, Sigma-Aldrich). Surface κ staining was done by incubating with a rat anti-mouse κ− FITCmAb (#MKAPPA01, Thermo Fisher Scientific, 10 μg/ml) in 30% rat serum at 25 °C for 2 h. The cells were then washed and stained with Fluorescein/Oregon Green Polyclonal Antibody, Alexa Fluor 488 (#A-11090, Thermo Fisher Scientific) for 30 min at 25 °C. For intracellular staining, the cells were then permeabilized with BD Cytofix/Cytoperm prior to incubation with goat anti-mouse λ -TRITC pAb (#1060-03, Southern Biotech, 4 μg/ml) in BD perm/wash solution containing 30% rat serum. After 2 h at 25 °C, the cells were washed and mounted on a slide with Fluoromount Aqueous Mounting Medium (#F4680, Sigma-Aldrich).

For the combined ER and λ staining, the cells were fixed and intracellular λ staining was done. Next, ER staining was performed using the ER Staining Kit - Green Fluorescence−Cytopainter (#ab139481,

Abcam). The mounted slides were imaged with Zeiss LSM880 (60X/1.42 Oil objective) microscope operating in the Airyscan mode and processed with the Zeiss Zen non-iterative deconvolution algorithm for Airyscan images. The images were cropped and contrast-enhanced for presentation purpose with Fiji (Image J).

## Primers

All primer sequences are in 5'→3'. For gene specific RT PCR, the primers described in ref. 66 were used: ACAGTCACTGAGCTGC (IgG) and AGTGGCCTCACAGGTAR (κ) to amplify mRNA. To amplify the H chain, forward primer (FP) MsV$_H$E (GGGAATTCGAGGTGCAGCTGCAG-GAGTCTGG), which binds all mouse V$_H$[67] was used in combination with reverse primers (RP) which bind the C region of the following isotypes: ATAGACAGATGGGGGTGTCGTTTTGGC (IgG1), CTTGACCAGGCATCC TAGAGTCA (IgG2a), AGGGGCCAGTGGATAGACTGATGG (IgG2b), AGG GACCAAGGGATAGACAGATGG (IgG3). For κ, mCκ (RP) GATGGTGGGA AGATGGATACAGTT that binds constant domain of all κ families was used together with mixture of FP that bind different Vκ families: GGTCAGACAGTCAGCAGT (Vκ1), GTGCTCTGGATTCGGGAA (Vκ2), TG CTGCTGCTCTGGGTTCCAG (Vκ3), ATTWTCAGCTTCCTGCTAATC (Vκ4),TTTTGCTTTTCTGGATTYCAG (Vκ5), TCGTGTTKCTSTGGTTGTC TG (Vκ6), ATGGAATCACAGRCYCWGGT (Vκ6,8,9), GTTGTAATGTCCA GAGGA (Vκ11), GCTTACAGGTGCCAGATGT (Vκ12/13), TCTTGTTGCT CTGGTTYCCAG (Vκ14), CAGTTCCTGGGGCTCTTGTTGTTC (Vκ19), CTSTGGTTGTCTGGTGTTGA (Vκ19/28), CTCACTAGCTCTTCTCCTC (Vκ20)[68], TGCTKCKCTGGGTTCCAG (Vκ21), TGTCTGGTGCCTGTGCA (Vκ22), TGGAYTYCAGCCTCCAGA (Vκ23), WTCTCTRGAGTCAGTGGG (Vκ24/25), GCTTCATGGTCAGTG (Vκ31/38), TGTTCTGCTTTTTAGGTG TG (Vκ32), GAATCCCAGGCATGATATGT (Vκ33/34), and GATATCAGG TGCCCAGTG (VκRF)[69]. For λ2[315] PCR, the FP (CAGGCTGTTGTGACT-CAG) that bind V region of λ1/2 and RP (GCCGAAAACAAAATGGT TTCTGAACC) that bind specifically to F[94]R[95]N[96] in λ2[315] and not germline λ2) was used.

## Preparation of cDNA library

Spleen cells from terminal Vλ2[315m+/−] Id-specific TCR-TG[+/−] mice were injected into Rag1[−/−] mice. Once the Rag1[−/−] recipients reached the humane endpoint, mice were culled, and single spleen cells were prepared. Approximately $5 \times 10^6$ cells were used to isolate RNA using Qiagen RNeasy kit. Subsequently, gene-specific cDNAs were made for Ig H and L chains with specific primers described above (for IgG, κ) using First strand cDNA Kit (#K1612, Thermo Fisher Scientific) and then PCRs using gene specific primers and Phusion Flash polymerase (#F548S, Thermo Fisher Scientific). The PCR products were purified using AMPURE beads. The samples were sequenced using MiSEQ Nano at the The Norwegian Sequencing Centre, Oslo University hospital, Oslo.

## Data processing

Raw sequencing files (.fastq.gz files, paired-end reads) were obtained from an Illumina Nano sequencing run. Reads were processed with the MiXCR software package[70] (version 4.3.20) including sequencing read alignment to a BALB/c mouse-specific germline gene database from OGRDB[36] followed by clonotype assembly using the V(D)J region sequence. Clonotypes were exported as tsv files for further analysis. SHM analysis was performed using MIXCR, which reported the number of mutations. Functional clonotypes (referring to unique sequences of VDJRegion(nt) after excluding out-of-frames and sequences containing stop codons) were included in the analysis only if they had a minimal read count of 2[71]. Charge was calculated using the R package peptides[72]. The alignment of the dominant sequence and the phylogenetic tree were made with CLC Main workbench (Version 8.1.3, QIAGEN Aarhus A/S). The aligned sequences were submitted to Genbank: accession numbers: PV068692 - PV068753 and PV068754 - PV068825.

## BCR recombinant immunoglobulin cloning and expression

The dominant $VDJ_H$ and $VJ_k$ sequences from four different lymphomas (NT7, NT8, NT18, NT24) were ordered from Genscript. $VDJ_H$ and $VJ_k$ were cloned into pLNOH2 (hγ3) and pLNOK (h$C_k$) vectors, respectively, using the signal peptide sequence from the $V_H$ gene of B1-8 mAb[73]. HEK293E (ATCC, Manassas, USA) cells cultured in multilayer cell culture flask (#353144, Falcon), were transiently transfected with combinations of H chain and κ L chain plasmids in a 1:1 ratio in the presence of Polyethylenimine (Polysciences). Supernatants were collected on days 4, 7 and 10. The recombinant immunoglobulins (rIgs) were purified on CaptureSelect FcXL affinity (#194328005, Life Technologies) using AKTA Prime Plus (GE HealthCare) equipment. The eluted Igs were dialysed against PBS and used in various assays. rIgs were detected by sandhich ELISA using mouse anti-human IgG ($C_H$3 domain, MA5-16557, Thermo Fisher Scientific, 1 μg/ml) and biotin anti-human L chain κ mAb (#316504, Biolegend, 1 μg/ml). The concentration of rIgs was measured on Nanodrop Lite Plus (Thermo Fisher Scientific). Anti-mouse CD40 rIg was made by cloning $VDJ_H$ and $VJ_k$ from anti-CD40 mAb (clone FGK45) into pLNOH2 (hγ3) and pLNOK (h$C_k$) plasmids. rIg purified from supernatants of transfectants was used as a negative control.

## Competition assay for H-L chain assembly

HEK293E cells cultured in 6-well plates were transfected with three plasmids: an H chain plasmid containing $V_H$ from the various lymphoma clones was fused to a hγ3 constant regions along with two L chain plasmids (lymphoma Vκ from the corresponding clone fused to a hCκ constant region and Vλ2[315] fused to a hCλ constant region from the M315) in 1:1:1 ratio. The supernatants were collected on day 4 and 7. The expression of κIg and λIg was detected using sandwich ELISA using mouse anti-human IgG (clone R10Z8E9, Invitrogen, #MA5-16929, 1 μg/ml) and biotin anti-human L chain κ mAb (clone MHK-49, Biolegend, #316504, 1 μg/ml) or biotin anti-human L chain λ mAb (clone MHL-38, Biolegend, #316604, 1 μg/ml). Biotinylated mAb was detected with streptavidin-Alkaline Phosphatase (ALP) conjugate (#RPN1234, Southern Biotech, 1:3000). Signals were developed with phosphatase substrate (#P4744, Sigma-Aldrich, 1 mg/ml) dissolved in diethanolamine substrate buffer. $OD_{405nm}$ was measured using EnVision 2104 Multilabel Reader with EnVision Manager 1.12 software (PerkinElmer).

## Assays to detect serum autoantibodies autoreactivity of rIg from lymphoma BCR

ANA HEp-2 Fluorescent Test System (#2040, Immunoconcepts) was used to detect autoantibodies as described[74]. Detection of serum antibodies by indirect immunofluorescence: Goat anti-mouse IgG (H +L) Alexa Fluor™ 488 (#A28175, Invitrogen, 1:1000), anti-mouse IgG1 (γ1) PE (#P21129, Invitrogen, 1:500), mouse IgG2a (γ2a) AF568 (#A21134, Invitrogen, 1:2000), anti-mouse IgG1 (γ1) AF488 (#A21121, Invitrogen, 1:1000), mouse IgG2a (γ2b) AF546 (#A21143, Invitrogen, 1:2000) and anti-mouse IgG1 (γ3) AF488 (#A21151, Invitrogen, 1:1000). Detection of recombinant Igs: FITC labeled anti-human IgG (from ANA kit) was used.

*Crithidia luciliae* immunofluorescence test (CLIFT) screening was done using nDNA Fluorescent Test System (#3040G, Immunoconcepts). After staining, the slides were mounted on coverslips using mounting media (from kit). The slides were stored at 4 °C protected from light overnight, before imaging with a Nikon Eclipse Ti fluorescent microscope (Nikon Instruments).

For histone ELISA, the 96-well high-binding EIA/RIA plates (Corning) were coated with 2 μg/ml whole histones (#HIS-1010, Immunovision Inc) diluted in PBS overnight at 4 °C. The mice serum samples and recombinant IgG were serially diluted in ELISA buffer (starting at respectively 1/200 and ¼) and incubated for 2 h at 37 °C. Anti-histone autoantibodies were detected with 1 μg/ml anti-mouse IgG HRP (#405306, Biolegend), biotinylated anti-mouse IgG1[a]

(#553500, BD Biosciences, 1:1000), anti-mouse IgG2a[a] (#553502, BD Biosciences, 1:1000), anti-mouse IgG2b[a] (#553393, BD Biosciences, 1:1000) or anti-mouse IgM[a] (#553515, BD Biosciences, 1:1000). Anti-nucleosome autoantibodies were detected using Anti-nucleosome ELISA (IgG) kit from Euroimmun (#EA 1574-9601). Sera and recombinant Ig were serially diluted in ELISA buffer (as above). Anti-nucleosome autoantibodies were revealed by anti-mouse IgG HRP. For the detection of antibodies against dsDNA, EliA dsDNA Well (precoated) (#14-5500-01, Phadia AB) was used. For the detection of autoantibodies against dsDNA in mouse serum, goat anti-mouse IgG HRP (#405306, Biolegend, 1:5000) was used.

To determine the specificity of BCR-derived rIg, immunoblot strips (#DL1590640136, EUROIMMUN) coated with parallel lines of highly purified self-antigens were used. The test strip was put into the incubation channel, and the sample buffer was added and incubated for 5 min with gentle shaking. 30 μl of recombinant Ig (about 0.1 mg/ml) was diluted in 1.5 ml sample buffer. The diluted samples were incubated for 30 min with gentle mixing, before strips were washed and incubated with enzyme conjugate for 30 min. After washing, strips were incubated with substrate for 10 min and washed. Scanning and evaluations of the blots were performed with EURO-LineScan (digital).

## Statistics

Two experimental groups were compared using two-tailed unpaired Student's *t*-test with Welch's correction, provided the samples had a normal distribution. A non-parametric two-tailed Mann–Whitney U test was used if there was no normal distribution. Several experimental groups with normal distribution were compared using one-way ANOVA and the Tukey/Dunnet's multiple comparisons test. Two-sided Kruskal–Wallis test, Dunn's multiple comparison was used when comparing experimental group with several control groups without normal distribution. Comparison of survival time lengths on the tumor avoidance curve was done using the Mantel–Cox test. GraphPad Prism (Version 8.4.0 and Version 10.2) was used to make plots and perform statistical analysis. All the plots were shown as scattered dot plot or violin plot, if the data set was large. Mean±SD was used to describe the distribution of biological replicates, while mean±SEM was applied when the objective was to convey the precision of the estimated mean. $^*p < 0.05$, $^{**}p < 0.01$, $^{***}p < 0.001$ and $^{****}p < 0.0001$.

## Reporting summary

Further information on research design is available in the Nature Portfolio Reporting Summary linked to this article.

## Data availability

The aligned lymphoma BCR sequences data generated in this study have been deposited in the Genbank database under accession code PV068692 - PV068753 [https://www.ncbi.nlm.nih.gov/nuccore/?term= VLambda2%2B%2F-+Id-sp+TCR-Tg%2B%2F-] and PV068754 - PV068825 [https://www.ncbi.nlm.nih.gov/nuccore/?term=Vlambda2315%2B%2F- +Id-sp.+TCR-TG%2B%2F-]. The mass spectrometry data from serum and recombinant Immunoglobulin data generated in this study have been deposited in Pride under Accession numberPXD063326 [https:// www.ebi.ac.uk/pride/archive/projects/PXD063326]. All data are included in the supplementary section. The raw numbers for charts and graphs are available in the Source Data file. Source data are provided with this paper.

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

## Acknowledgements

We would like to thank Peter Olaf Hofgaard and Hilde Omholt for their technical assistance. Thanks to Anders Tveita for his assistance in growing lymphoma cells in vitro. Ida Jonsson helped with viral transfection of NT34 lymphoma cell line with Effluc.GFP. We would like to thank the late Herbert C. Morse III for initiating the collaboration on lymphoma diagnosis. We acknowledge the staff of the Department of Comparative Medicine at Oslo University Hospital for their support and assistance with the animal housing facilities. We also acknowledge the proteomic core facility at Oslo University Hospital (supported by INFRASTRUKTUR-program (project number: 295910)) and Tuula Nyman and Sachin Singh for their help with mass spectrometry analysis. Prof. Oddmund Bakke is thanked for advice on confocal microscopy. We would also like to thank the following funding agencies: The Research Council of Norway: projects 221709 and 143073 (to BB); The Regional Health Authority (Helse Sør Øst): projects 40150 and 39921 (to BB).

## Author contributions

Designed experiments: R.P.G, P.C.H and B.B; performed experiments: R.P.G, P.C.H, J.M.W, V.G, R.B, X.H, L.B, K.L.Q, L.M. Analyzed experiments: All authors. Wrote the paper: R.P.G and B.B. Correspondence and material requests should be directed to Prof. Bjarne Bogen (bjarne.bogen@medisin.uio.no). All authors read and approved the final manuscript.

## Competing interests

The authors declare no competing interests.
