## [Transparent Peer Review file · Nature Communications]

Idiotype-specific CD4+ T cells chronically stimulate autoreactive B cells to develop into B lymphomas in mice

Corresponding Author: Dr Ramakrishna Prabhu Gopalakrishnan

A version of this paper was originally rejected for publication by Nature Communications, however that decision was reconsidered after appeal by the authors.

Version 0:

Reviewer comments:

Reviewer #1

(Remarks to the Author)

In this manuscript, a novel mouse model is presented that may explain the unresolved question why autoimmune diseases are often linked to the development of B cell lymphomas. The model is based on an F1 backcross of two transgenic mouse strains. One strain expresses in approximately 0.5% of all B cells a hypermutated Ig lambda light chain (derived from a plasmacytoma mouse line). By pairing with any endogenous heavy chain, a few B cells might recognize autoantigens. The second mouse line expresses a transgenic T cell receptor specific for the hypermutated motif in the lambda light chain (anti-idotype TCR). Thus, these T cells (which surprisingly don't get clonally deleted in the F1 strain) can provide help to the B cells independent of the specificity of the BCR (Id-driven T-B collaboration). F1 mice develop overt autoimmunity and also B cell lymphomas from the age of 12 months.

This is a very interesting and also clinically highly relevant mouse model since it could nicely explain why several autoimmune diseases, such as Sjogren's syndrome, are associated with a high risk of lymphoma development.

Specific comments

1. The group of Bjarne Bogen, who has been working on the question of Id-driven T-B collaboration for more than 30 years, previously published a similar mouse model (Zangani et al., J Exp Med, 2007). In that model, all B cells expressed the transgenic light chain and T cells were adoptively transferred. Similar to the new model, mice developed autoimmunity as well as lymphomas. While the new light chain transgenic strain with only about 0.5% expressing cells is much more physiological and a clear improvement, it is not clear why the authors switched back to an F1 backcross with the TCR transgenic strain instead of adoptively transferring of a limited number of autoreactive T cells, which would be much more physiological and also avoid potential artifacts of thymic selection. It would be good at least to see whether in such a setup some of the rather unexpected observations made in this study (such as the lambda/kappa double expressing B cells) still exist.
2. The flow cytometry results of all intranuclear stains (FoxP3, Ki-67) look really strange with no clearly separated positive population. According to the Methods section, the BD Cytotfix/Cytoperm (BD #554714) kit was used. This kit is designed for cytoplasmic, not intranuclear staining! Experiments have to be repeated using the appropriate kit (BD Pharmingen™ Transcription Factor Buffer Set, # 562574).
3. Information on number of independent experiments and animals per group is missing for all experiments.
4. For statistical analysis, Student's t-test or one-way ANOVA were used throughout all experiments. For many experiments this is inappropriate as there is obviously no normal distribution (e.g. Fig. 1D, 2D, 2G and many more). For many experiments, also the group sizes with sometimes as few as three animals are too small (after removing one outlier, any trends for differences between groups are gone, e.g. Fig. 2 E).
5. Figure 3 M: According to the text and Fig. 3L, kappa expression in the B lymphoma group is reduced. The graph shows

the opposite.

6. Figure 3 N: Please check compensation.

7. Suppl. Fig. S1 B: How do you explain the reduced TCR Vb8 expression in the Id. sp TCR+/- group compared to the double transgenic mouse?

8. Suppl. Fig. S3 K: Please check compensation; the sharp peak obviously results from cells piling up in the sum channel of another fluorophore.

Reviewer #2

(Remarks to the Author)

In the study by Gopalakrishnan and colleagues, a mouse model was established to study whether idiotypic-specific CD4+ T cells promote the development of idiotypic+ autoreactive B cells into lymphomas. Prior studies by the group of the senior author and by other groups already showed that principally, idiotypic-specific CD4+ T cells can be generated for mutated B cell receptors (BCRs), and there is a known link between autoimmunity and the development of B cell lymphomas. So the rationale is clear and a highly relevant question is addressed.

The model was meant to be more physiological than standard transgenic mouse models, because only a small number of B cells will express the idiotypic (only B cells which during their development rearrange a mutated Vlambda2 gene to the Jlambda2 gene), and not nearly all B cells. The hypothesis for the model was that if a BCR using the Vlambda2 light chain is by chance autoreactive, the autoreactive B cells would be stimulated, upregulate MHC class 2 and present the light chain idiotypic to the transgenic idiotypic-specific CD4+ T cells, which then further stimulate the autoreactive B cells, drive their constitutive clonal expansion and finally lead to lymphoma development. The mice indeed developed autoantibodies and after about a year in some mice B cell lymphomas developed (other developed T cell lymphomas). Hence, at first glance these findings seem to nicely support the hypothesis of the study. However, a closer inspection of the mice and their B cells revealed that the autoantibodies expressed kappa light chains, and also the lymphomas expressed a kappa BCR. The lymphomas nevertheless developed from Vlambda2 idiotypic-positive B cells, but the lambda light chains were only detected within the cells, and not as part of the BCR. Thus, allelic exclusion is violated here and the B cells co-express a kappa and a lambda light chain (but only one as part of the BCR). In the lymphomas, the BCR with the kappa light chain provides the autoreactivity, whereas peptides from the lambda light chain are presented on MHC class 2 and thereby mediate stimulation of the B cells through the idiotypic-specific CD4+ T cells. The reason for the violation of allelic light chain exclusion is not clear. Perhaps, the mutated Vlambda 2 gene has problems to pair stably with most Ig heavy chains, so that additional light chain rearrangements are performed, leading to allelic inclusion. Dual light chain producing B cells have been identified, but this is clearly not the general situation in autoimmune diseases and B cell lymphomas.

The experiments are principally well done and nicely presented.

So in my opinion the study presents an unexpected and curious observation, but neither validates nor falsifies the hypothesis of the study. The original idea that an idiotypic peptide of an autoreactive BCR is presented on MHC class 2 and causes chronic stimulation of the B cells through idiotypic-specific CD4+ T cells, finally leading to lymphoma development, is not tested here. This is because in the present model, unexpectedly not an idiotypic peptide from the autoreactive BCR is presented to the CD4+ T cells, but a peptide from another protein, that is not part of the BCR (i.e., the lambda light chain).

Minor point

a) From line 106 onwards there are multiple instances where the mouse line designation is wrongly written. It should be V12 and then "315m+/-" in superscript, and not V1 with "2315m+/-" in superscript.

b) Is there a mistake in figure 3M? According to the text the double transgenic mice (red symbols) have lower kappa light chain expression than the controls, but here the controls all have apparently no kappa light chain expression, but the double transgenic B cells show variable expression. Please clarify.

Reviewer #3

(Remarks to the Author)

Reviewer #4

(Remarks to the Author)

Gopalakrishnan et al. have attempted to pursue the links between autoimmunity and lymphomagenesis. They created a model wherein they generated mice with a modified I light chain creating an epitope that binds to I-Ad. These mice were crossed into TCR transgenic mice that recognize the I-Ad peptide combination, generating mice heterozygous both for the light chain epitope and for the TCR transgene. These mice develop some anti-nuclear antibodies and eventually develop lymphomas. The lymphomas are kappa light chain expressors and H-chain sequences were shared between earlier

autoantibody BCRs and lymphoma BCRs.

While the approach has potential, and this work has some interesting aspects the main issue in the manuscript appears to be an apparent disregard for some fundamental immunological principles and the authors' apparent decision to ignore their own data and not try to interpret it. The fundamental premise in the schematic in Figures 1A and B is not based on a sound understanding of the principles of T-B collaboration and is fundamentally incorrect, based on the data that the authors present, but inexplicably choose not to interpret and end up ignoring. A key experiment to test the likely flaws in the model in the schematic was ignored. This will be pointed out below

1. Surprise is repeatedly expressed that the lymphomas were not lambda light chain positive! The authors show in Figures 1 panels H, I and J that the anti-nuclear antibodies they observe are almost exclusively kappa light chain positive. 90% of all Igs in mice are kappa positive and in the obvious scenario painted below it would be likely that most responding B cells would be kappa positive.
2. The likely simple explanation of the data is that some IgHlambda2315 immunoglobulin molecule/s which are essentially provided in relatively high doses in these mice, are recognized by a self-reactive BCR or BCRs (at least 60-70% of all BCRs are self-reactive in practical terms) on one or a few B cell clones all of which are kappa expressors (not surprising - there is a 90% chance that a BCR would be kappa positive). Some (possibly a small fraction) of these BCRs in these mice are also capable of recognizing some epitope on a nuclear antigen.
3. These kappa+ self-reactive BCRs internalize the IgHlambda2315 immunoglobulin self-antigen, process it and present the lambda peptide to the relatively large numbers of CD4+ T cells that are specific for the peptide. The kappa + self-reactive B cell and the CD4+ T cell collaborate at the T-B boundary generate an extrafollicular B cell response that generates plasmablasts secreting anti-nuclear antibodies. No autoimmune disease necessarily, just an autoimmune phenomenon which is not such a big deal
4. As these kappa+ self-reactive B cell clones are repeatedly stimulated by T cell help since antigen is always available, and therefore expand, occasionally additional hits accumulate and lymphomas evolve. The data says most of this!!!
5. The authors should do the following:
 - a) use the Wardemann and Nussenzweig approach and clone V regions of Ig H and kappa light chains from some lymphomas, clone into the already widely used Ig vectors, and then express in 293T cells
 - b) show that these clonal Igs are self-reactive using the assays they have used in the manuscript
 - c) delete and correct the old schematic in Figure 1
 - d) In the abstract and manuscript stick to the use of "autoantibodies" or "autoimmune phenomena" and delete "autoimmunity" in the manuscript. There is no good evidence for an autoimmune disease in this manuscript. We all have autoantibodies! Self-reactivity is a normal phenomenon – some extreme self-reactivity is eliminated by central tolerance mechanisms.

Version 1:

Reviewer comments:

Reviewer #1

(Remarks to the Author)

The revised version of the manuscript satisfactorily answered most of the open questions. Only the intranuclear stainings shown in the manuscript require further attention.

1. Please compare the FoxP3 staining shown in Suppl. Fig. S1D to a typical outcome shown on the manufacturer's homepage (clone FJK-16s from Thermo Fisher Scientific; not "FJK-162 by Thermofischer" - see below). What you expect is a clearly separated positive population, not a shift of the entire population and a high background over the whole fluorescence range. To demonstrate specificity of the staining, the authors should plot FoxP3 against CD4 (gated on CD3+ or live) and FoxP3 against TCRb8 or GB113 (gated on CD4+). This would be a better way than the isotype control to judge signal versus background.

2. This manuscript requires careful reading by the senior corresponding author - there are so many inconsistencies and small errors! Please check clone names (see above), spelling of fluorophores (e.g. it is not "PecP-vio700" but "PerCP-Vio700") and manufacturer's names (e.g. it is not "Miltney BioTech" but "Miltenyi Biotec").

Reviewer #2

(Remarks to the Author)

My criticism to the first version of the manuscript has been adequately addressed. The manuscript is further improved in response to issues raised by the other referees.

There is one minor point:

In the legend to figure 7, there is a mixup of letters for the subfigures O -Q, and N is missing.

Reviewer #3

(Remarks to the Author)

Reviewer #4

(Remarks to the Author)

All issues have been addressed.

Version 2:

Reviewer comments:

Reviewer #1

(Remarks to the Author)

All issues have been addressed

Rebuttal letter to the Reviewers' comments:

Reviewer #1 (Remarks to the Author):

In this manuscript, a novel mouse model is presented that may explain the unresolved question why autoimmune diseases are often linked to the development of B cell lymphomas. The model is based on an F1 backcross of two transgenic mouse strains. One strain expresses in approximately 0.5% of all B cells a hypermutated Ig lambda light chain (derived from a plasmacytoma mouse line). By pairing with any endogenous heavy chain, a few B cells might recognize autoantigens. The second mouse line expresses a transgenic T cell receptor specific for the hypermutated motif in the lambda light chain (anti-ideotype TCR). Thus, these T cells (which surprisingly don't get clonally deleted in the F1 strain) can provide help to the B cells independent of the specificity of the BCR (Id-driven T-B collaboration). F1 mice develop overt autoimmunity and also B cell lymphomas from the age of 12 months.

This is a very interesting and also clinically highly relevant mouse model since it could nicely explain why several autoimmune diseases, such as Sjogren's syndrome, are associated with a high risk of lymphoma development.

***Our response:** We would like to thank the Reviewer for highlighting the novelty of our work and finding our mouse model to be interesting and clinically highly relevant.*

Specific comments

Comment 1. The group of Bjarne Bogen, who has been working on the question of Id-driven T-B collaboration for more than 30 years, previously published a similar mouse model (Zangani et al., J Exp Med, 2007). In that model, all B cells expressed the transgenic light chain and T cells were adoptively transferred. Similar to the new model, mice developed autoimmunity as well as lymphomas. While the new light chain transgenic strain with only about 0.5% expressing cells is much more physiological and a clear improvement, it is not clear why the authors switched back to an F1 backcross with the TCR transgenic strain instead of adoptively transferring of a limited number of autoreactive T cells, which would be much more physiological and also avoid potential artifacts of thymic selection. It would be good at least to see whether in such a setup some of the rather unexpected observations made in this study (such as the lambda/kappa double expressing B cells) still exist.

Our response:

The Reviewer is right in stating that: "...the new light chain transgenic strain with only about 0.5% expressing cells is much more physiological and a clear improvement,..." . In a further comment to the Zangani paper, the Reviewer writes: " Similar to the new model, mice developed autoimmunity as well as lymphomas." . However, the Zangani et al. paper did not study autoimmunity at all. Thus, the link between autoimmunity and later B lymphoma

development is novel to the present study and represents a major advance, as indicated by the title of the present paper.

The Reviewer wonders why we in the present study use F1 backcross mice rather than transfer of T cells. In our opinion, it is more physiological not to transfer T cells (which had to be done multiple times in the Zangani paper, in addition, that study used unphysiological in vitro-polarized Th2 cells). In contrast, in the present study, naïve Id-specific T cells develop along normal pathways, including thymic education, which we feel is more physiological.

The Reviewer asks why we have not transferred limited numbers of autoreactive (idiotype-reactive?- our comment) T cells into V gene-modified mice. This is a relevant question since Id-specific TCR-transgenic mice have an unphysiologically high frequency of Id-specific CD4⁺ T cells. We already did the experiment requested by the reviewer, but did not include the results in the submitted manuscript. The data have now been included in the revised manuscript. In brief, 6 Vλ2^{315m+/-} mice (2-month-old) were transferred with 1 million naïve Id-specific CD4⁺ T cells. Controls were 4 age-matched Vλ2^{315m+/-} mice that did not receive any T cells. Within 13 months, 2/6 of transferred mice developed autoantibodies. Later these two mice developed B cell lymphomas. Another 2 transferred mice developed B cell lymphomas without preceding autoimmunity. Thus, altogether 4 out of 6 transferred mice developed B cell lymphomas within 20 months. In lymphoma mice, there was expansion of Id-specific T cells and pId:MHCII⁺ B cells. None of the control mice (0/4) developed autoantibodies or lymphomas. This experiment shows that even when strongly reducing the number of Id-specific T cells, and circumventing T cell development in the thymus, autoimmunity and B lymphomas were induced albeit at lower levels. The data obtained in this transfer experiment has not allow us to address the issue of lambda/kappa double-expressing B cells. A difficulty has been that we don't have adequately frozen cells to establish secondary lymphomas in Rag1^{-/-} recipients.

Changes done in the revised manuscript: *The transfer experiment asked for by the reviewer is presented in Fig. 7M-Q, Supplementary Fig. S8 and Supplementary Table-1. The experiments are described in the Methods (line 502-507), Results (line 344-361) and in the Discussion (lines 446-451).*

Comment 2. *The flow cytometry results of all intranuclear stains (FoxP3, Ki-67) look really strange with no clearly separated positive population. According to the Methods section, the BD Cytofix/Cytoperm (BD #554714) kit was used. This kit is designed for cytoplasmic, not intranuclear staining! Experiments have to be repeated using the appropriate kit (BD Pharmingen™ Transcription Factor Buffer Set, # 562574).*

Our response: *We have repeated the FOXP3 staining (Fig. 1E) with BD Pharmingen™ Transcription Factor Buffer Set, # 562574 and have updated the figure and Materials and Methods. Regarding the nuclear antigen Ki-67 staining (Fig. 3E and 4E), it was done as described in Graefe C et al., Optimized Ki-67 staining in murine cells: a tool to determine cell proliferation. Mol Biol Rep. 2019.*

Changes done in the revised manuscript: We have updated the FoxP3 staining (Fig 1E). We also updated the Materials and Methods section to explain the Ki-67 staining in more detail, and also referred to the Graefe et al. paper mentioned above (lines 562-565).

Comment 3. Information on number of independent experiments and animals per group is missing for all experiments.

Our response: We thank the reviewer for pointing this out.

Changes done in a revised manuscript: Information on number of independent experiments, and mice per group is now stated in the legends of Figs. and Supplementary Figs. These changes are too many to be indicated individually but changes are clearly marked in the revised manuscript.

Comment 4. For statistical analysis, Student's t-test or one-way ANOVA were used throughout all experiments. For many experiments this is inappropriate as there is obviously no normal distribution (e.g. Fig. 1D, 2D, 2G and many more). For many experiments, also the group sizes with sometimes as few as three animals are too small (after removing one outlier, any trends for differences between groups are gone, e.g. Fig. 2 E).

Our response: We thank the reviewer for pointing this out.

Concerning statistics: We have repeated the statistics according to the Reviewers' suggestion. Two experimental groups were compared using unpaired Student's T-test with Welch's correction provided the samples had a normal distribution. A non-parametric Mann-Whitney U test was used, if there was no normal distribution. Several experimental groups, with normal distribution, were compared using one-way ANOVA or the Tukey's multiple comparisons test. Kruskal-Wallis test, Dunn's multiple comparison was used when comparing several experimental groups not having normal distribution. Survival curves were compared using the Mantel-Cox test.

When we repeated the statistical analysis as described above, most of the significant p values were retained. However, in two cases, Fig. 2C and 2E, the significance was lost. These two Fig. items have been transferred to Supplementary Figs. as new Fig. S2D and Fig. S2H. The Results text has been correspondingly changed, emphasizing that the results in Fig. S2D and Fig. S2H are trending but not statistically significant.

Concerning group sizes: The group sizes in the various experiments of the originally submitted paper are given below.

Fig 1: C,D: 3-4 mice/group; E: 3-4/group; G: 7/group; I: 6-15/group; J: 6; K: 6-12/group; L: 4-14/group; M: 14; N: 4-11/group; O: 12, P: 11. **Fig. 2:** A-C: 4/exp. group, 3/control group; D, E, F, G: 4-5/group; H: 15-40/group; I: 4-14/group; J: 7-12/group; M: 40 (total number of classified lymphomas). **Fig. 3:** A-E: 10-12/group; F: 10/group; G: 12-14/group; I: 6/group; M: 8-10/group. **Fig. 4:** C:11-14/group; D-G: 6-13/group; L, M: 4/group. **Fig. 7:** F: 4/group, H- K: 4/group; M-Q: 6/exp. group and 4/control group. **Fig. S1:** A: 4-5/group; F: 4-5/group; G: 6/group; H, I: 6-15/group; Q: 14/ exp. group and 4-5/ controls groups. **Fig. S2:** A: 3-4/group; E-F: 3-4/group; I, J: 3-4/group; S2K: 12-14/cohort; L: 8; M-P: 3-8/group. **Fig. S3:** F-H, M: 4/group. **Fig. S8:** B-C: 6/ exp. group and 4/ control group.

As can be seen, we have generally used a sufficient number of mice/group for statistical comparisons except in Fig. 1C, D, E where only 3-4 mice/group were used. To remedy this, we have for Fig. 1C,D combined three similar experiments, each giving similar results, thereby increasing group sizes to 6-10(11) mice/group. For Fig. 1E, we pooled data from two novel and independent experiments performed with BD Pharmingen™ Transcription Factor Buffer Set, # 562574 as recommended by the Reviewer (see Comment 2, above). Group sizes are now n=9/group.

The cytokine measurements in sera of 10-month-old mice (Fig 2F-G in submitted version) are in the cases of IL-10 and TNF- α somewhat weak (while the results for IL-21 and IFN γ are clear-cut). These data have now been bolstered by including cytokine levels in sera of 14-20 months old mice with B cell lymphomas (Fig. 2D-G in revised version). The novel data show that the two cytokines which were only weakly upregulated at 10 months (IL-10, TNF- α) were strongly upregulated with lymphoma development.

Changes done in the revised manuscript:

The statistics section in the Materials and Methods have been updated according to the Reviewer's suggestion (lines 804-812).

The statistical methods employed are described in the Fig. legends. Specifically, the statistical analysis has been updated for Fig. 3A, 3D, 3G, 4D, and 4G with maintenance of significance.

Fig. 1M, S2E, S2I and S2J lacked description of statistics in the submitted version were updated now.

Data of Fig. 2C and 2E became non-significant upon reanalysis, these two Fig items have been transferred to Supplementary as Figs. S2D and S2H, respectively.

The Results section have been altered to accommodate the changes (lines 138-143).

The number of mice in Fig 1C, D, E has been increased to 6-11 mice.

The cytokine data in Fig. 2D-G have been updated in the revised manuscript by including data from 14–20-month-old lymphoma mice. The corresponding text in the Results section have been updated (lines 150-155).

Comment 5. Figure 3 M: According to the text and Fig. 3L, kappa expression in the B lymphoma group is reduced. The graph shows the opposite.

Our response: *Thanks for pointing this out. We have mistakenly copied graph 3I onto 3M (i.e., duplicated 3I). How this inadvertently happened we do not know.*

Changes done in the revised manuscript: *The correct graph has been included in the revised manuscript. The new graph clearly illustrates the reduced k expression on B lymphomas compared to controls.*

Comment 6. Figure 3 N: Please check compensation.

Our response: We reanalyzed the data and checked the compensation. The compensation appears correct. We think the reason for the unusual appearance might be due to stretching of the axes of the contour plot which results in an impression of potential compensation issues.

Changes done in the revised manuscript: We have updated the contour plot without having stretched the axes.

Comment 7. Suppl. Fig. S1 B: How do you explain the reduced TCR Vb8 expression in the Id. sp TCR^{+/-} group compared to the double transgenic mouse?

Our response: Thank you for making this keen observation. Fig. S1B is related to Fig. 1C. As stated in our response to Comment 4 of the Reviewer, (above), we have combined 3 similar experiments in the new Fig. 1 C. Upon closer examination of the stainings from these three experiments, we don't think that there is a much-reduced TCR Vb8 expression in the Id-specific TCR^{+/-} group.

Changes done in the revised manuscript: We have replaced Fig. S1 B with a more representative histogram for the Id-specific TCR^{+/-} group.

Comment 8. Suppl. Fig. S3 K: Please check compensation; the sharp peak obviously results from cells piling up in the sum channel of another fluorophore.

Our response: Other histograms representing the same staining do not show this sharp peak.

Changes done in the revised manuscript: We replaced Fig. S3K with another representative histogram (without a sharp peak).

Reviewer #2 (Remarks to the Author):

In the study by Gopalakrishnan and colleagues, a mouse model was established to study whether idiotype-specific CD4⁺ T cells promote the development of idiotype⁺ autoreactive B cells into lymphomas. Prior studies by the group of the senior author and by other groups already showed that principally, idiotype-specific CD4⁺ T cells can be generated for mutated B cell receptors (BCRs), and there is a known link between autoimmunity and the development of B cell lymphomas. So, the rationale is clear and a highly relevant question is addressed.

The model was meant to be more physiological than standard transgenic mouse models, because only a small number of B cells will express the idiotype (only B cells which during their development rearrange a mutated V_{lambda}2 gene to the J_{lambda}2 gene), and not nearly all B cells. The hypothesis for the model was that if a BCR using the V_{lambda}2 light chain is by chance autoreactive, the autoreactive B cells would be stimulated, upregulate MHC class 2 and present the light chain idiotype to the transgenic idiotype-specific CD4⁺ T cells, which then further stimulate the autoreactive B cells, drive their constitutive clonal expansion and finally lead to lymphomas development. The mice indeed developed autoantibodies and after

about a year in some mice B cell lymphomas developed (other developed T cell lymphomas). Hence, at first glance these findings seem to nicely support the hypothesis of the study. However, a closer inspection of the mice and their B cells revealed that the autoantibodies expressed kappa light chains, and also the lymphomas expressed a kappa BCR. The lymphomas nevertheless developed from Vlambda2 idiotype-positive B cells, but the lambda light chains were only detected within the cells, and not as part of the BCR. Thus, allelic exclusion is violated here and the B cells co-express a kappa and a lambda light chain (but only one as part of the BCR). In the lymphomas, the BCR with the kappa light chain provides the autoreactivity, whereas peptides from the lambda light chain are presented on MHC class 2 and thereby mediate stimulation of the B cells through the idiotype-specific CD4+ T cells. The reason for the violation of allelic light chain exclusion is not clear. Perhaps, the mutated Vlambda 2 gene has problems to pair stably with most Ig heavy chains, so that additional light chain rearrangements are performed, leading to allelic inclusion. Dual light chain producing B cells have been identified, but this is clearly not the general situation in autoimmune diseases and B cell lymphomas.

Our response:

The reviewer nicely summarizes the results.

Concerning allelic inclusion, we agree with the possible explanation suggested by the Reviewer: “Perhaps, the mutated Vlambda 2 gene has problems to pair stably with most Ig heavy chains, so that additional light chain rearrangements are performed, leading to allelic inclusion”. Supporting the Reviewer’s view, we do present evidence that the various lymphoma BCR H chains poorly associated with $\lambda 2^{315}$ compared with the B lymphoma κ (Fig. 5F). In contrast, the H chain of the MOPC315 myeloma pairs well with $\lambda 2^{315}$ (Fig. 5F). We comment upon these data in Results, lines 243-254.

The Reviewer writes: “Dual light chain producing B cells have been identified, but this is clearly not the general situation in autoimmune diseases and B cell lymphomas.” Dual L chain B cells have been described in autoimmune diseases in both mouse and humans (references 50-53 in the manuscript). Similarly, human B cell malignancies with dual L chains have also been reported previously, mostly in case reports (Aggressive B lymphoma, Fujiwara T et al., Intern Med. 2007; Mantle cell lymphoma, Wang et al., Int J Lab Hematol. 2023; B-CLL, del Senno et al., Leuk Res. 1987 and Xu et al, Arch Pathol Lab Med. 2006). In a larger study on B-CLL patients, 1 out of 56 patients expressed dual L chains (Peltomaki et al., Eur J Cancer Clin Oncol. 1988). In all these studies, dual L chains were expressed in the cell membrane of malignant B cells as detected by flow cytometry. The frequency of dual L chain expression in human malignancies has not been exactly defined, but it is likely to be low, as the Reviewer suggests (e.g. in the 1-3% range?). However, and importantly, it could well be that the prevalence of dual L chain expression in B cell cancers could turn out to be higher if surface and intracellular staining is combined, similar to our findings in the current mouse model. This should be explored in future studies.

Changes done in the revised manuscript: *We already have 4 references (refs 50-53) related to dual L chain B cells in autoimmune diseases in mice and humans. We are hesitant to*

include more refs (see above) since the paper is already long. However, we include two new paragraphs on dual L-chain expression and receptor revision in the Discussion. This is described in more detail below.

The experiments are principally well done and nicely presented.

Our response: *We thank the reviewer for pointing this out.*

So in my opinion the study presents an unexpected and curious observation, but neither validates nor falsifies the hypothesis of the study. The original idea that an idiotypic peptide of an autoreactive BCR is presented on MHC class 2 and causes chronic stimulation of the B cells through idiotypic-specific CD4⁺ T cells, finally leading to lymphoma development, is not tested here. This is because in the present model, unexpectedly not an idiotypic peptide from the autoreactive BCR is presented to the CD4⁺ T cells, but a peptide from another protein, that is not part of the BCR (i.e., the lambda light chain).

Our response: *We appreciate the Reviewer's comment. We started with a hypothesis (Fig 1A) but we found something "unexpected and curious" as pointed by the Reviewer. In our opinion this makes the present work even more interesting. Clearly, the statement that "Id-driven collaboration can drive development of autoimmunity and B lymphoma" is still true. It is just that the Id-peptide is derived from an intracellular L chain, rather than a BCR L chain.*

Importantly, directly addressing the Reviewer's concern, we have recently discovered, in an aged V λ 2^{315m+/-} Id-specific TCR-TG^{+/-} mouse, a B lymphoma that expressed a λ 2^{315m+/-} BCR and that stimulates Id-specific CD4⁺ T cells. In the pre-lymphoma phase, the mouse had λ 2⁺ autoantibodies that bound nucleosome/histone. These novel findings directly supports the hypothesis of Fig. 1A. Interestingly, this B lymphoma has an intracellular κ and is likely to have undergone a $\kappa \rightarrow \lambda$ 2^{315m} revision.

Based on the present results, and the previous findings that (i) BCR-ligation by antigen results in pId:MHCII presentation by normal B cells (Huszthy et al., PNAS 2019) and (ii) that nascent λ 2 L chains in the ER are antigen-processed and MHCII-presented (Weiss and Bogen Cell 1991), we consider it likely that the Id-peptide in the current B lymphomas can be derived from a λ 2 L chain that is either part of cell surface BCR or nascent λ 2 L chain in the ER.

It is striking that 23/24 (96%) of B lymphomas probably have undergone a λ 2^{315m} \rightarrow κ revision while 1/24 (4%) is likely to have undergone the reverse revision, $\kappa \rightarrow \lambda$ 2^{315m}. These frequencies correspond roughly to κ/λ ratios in mouse serum Ig. The H chain that happens to be expressed early in a B cell's development could influence this result by preferentially pairing with either κ or λ L chains. It is also striking that we did not find any B lymphomas that only expressed one L-chain, i.e. λ 2^{315m}. We speculate that this could be related to the reinstated Rag expression in dual L chain B cells that somehow could be lymphomagenic.

Changes done in the revised manuscript: For increased clarity, we added a new Fig 6M to the revised manuscript illustrating that the Id-peptide in the great majority of the present B cell lymphomas is derived from an intracellular source. The novel data on the $\lambda 2^{315m+}$ B lymphoma are presented in Fig. 7 A-L, Supplementary Fig. 7, and Results lines 313-343. Discussion lines 415-437 have been modified to encompass the novel data and a possible explanation for the results (see above). The new Fig. 7L illustrates that the $\lambda 2^{315m+}$ B lymphoma (and $\lambda 2^+$ autoantibodies) directly supports the hypothesis of Fig. 1A.

Minor point a)

From line 106 onwards there are multiple instances where the mouse line designation is wrongly written. It should be V12 and then "315m+/-" in superscript, and not V1 with "2315m+/-" in superscript.

Our response: Thanks for pointing this out.

Changes done in the revised manuscript: This has been changed from line 106-160.

Minor point b)

Is there a mistake in figure 3M? According to the text the double transgenic mice (red symbols) have lower kappa light chain expression than the controls, but here the controls all have apparently no kappa light chain expression, but the double transgenic B cells show variable expression. Please clarify.

Our response: Thanks for pointing this out. Reviewer 1 made the same observation. We have mistakenly copied graph 3I onto 3M (i.e., duplicated 3I). How this inadvertently happened we do not know.

Changes done in the revised manuscript: The correct graph has been included in the revised manuscript. The new graph clearly illustrates the reduced k expression on B lymphoma cells compared to controls.

Reviewer #3 (Remarks to the Author):

Our response: We would like to thank Reviewer #3 for reviewing this manuscript.

Reviewer #4 (Remarks to the Author):

Gopalakrishnan et al. have attempted to pursue the links between autoimmunity and

lymphomagenesis. The created a model wherein they generated mice with a modified I light chain creating an epitope that binds to I-Ad. These mice were crossed into TCR transgenic mice the recognize the I-Ad peptide combination, generating mice heterozygous both for the light chain epitope and for the TCR transgene. These mice develop some anti-nuclear antibodies and eventually develop lymphomas. The lymphomas are kappa light chain expressors and H-chain sequences were shared between earlier autoantibody BCRs and lymphoma BCRs.

Our response: *Just a minor point: the Id epitope binds I-E^d, not I-A^d.*

While the approach has potential, and this work has some interesting aspects the main issue in the manuscript appears to be an apparent disregard for some fundamental immunological principles and the authors' apparent decision to ignore their own data and not try to interpret it. The fundamental premise in the schematic in Figures 1A and B is not based on a sound understanding of the principles of T-B collaboration and is fundamentally incorrect, based on the data that the authors present, but inexplicably choose not to interpret and end up ignoring. A key experiment to test the likely flaws in the model in the schematic was ignored. This will be pointed out below

Our response: *With all due respect, the Reviewer appears to have overlooked crucial information in the manuscript which has led to a misunderstanding on part of the Reviewer. The paper is not about conventional T-B collaboration (Lanzavecchia Nature 1985), as reviewer 4 appears to think, but rather about Id-driven T-B. The concept of Id-driven T-B collaboration is described in detail in the Introduction (lines 37-69, refs 9-25) and in Fig. 1A, the concept has been established by us and others over the last 30 years. In contrast to Reviewer 4, Reviewers 1-3 have perfectly well understood Id-driven T-B collaboration and the hypothesis of Fig. 1A. Thus, it should not be anything wrong with the description of Id-driven T-B collaboration in the Introduction, and the hypothesis in Fig 1A. However, Id-driven T-B collaboration is clearly less well known compared to conventional T-B collaboration which probably contributes to the misunderstanding.*

This having been said, soluble Id⁺ Ig can induce conventional T-B collaboration where B cells express an anti-Id BCR that bind Id+ Ig and where pId:MHCII is recognized by Id-specific CD4⁺ T cells. This was shown by senior author Bogen and colleagues by generating and employing an Ig knock-in mouse that expresses an anti-Id BCR (Jacobsen et al. Naive idiotope-specific B and T cells collaborate efficiently in the absence of dendritic cells. J Immunol. 2014). Reviewer 4 is obviously thinking along these lines as becomes clear from the Reviewer's comment further below. But Id in conventional T-B collaboration is not the topic of the present paper; rather, the topic is Id-driven T-B collaboration as outlined in the Introduction and Fig. 1A.

Changes to be done in the revised manuscript: *Since Reviewer 4 has misunderstood crucial aspects of the paper, it is important to prevent other readers from also misunderstanding. To this end, a sentence in the Introduction (line 42) has been modified. It now reads: "This phenomenon is the basis for the concept of Id-driven T-B collaboration, which is distinct from*

conventional T-B collaboration^{11,12}". Ref 11 (Lanzavecchia Nature 1985) is new, ref 12 (Muthe et al. J Immunol 2004) was included in the submitted version but has now been moved and got a new number.

Comments 1 and 2.

1. Surprise is repeatedly expressed that the lymphomas were not lambda light chain positive! The authors show in Figures I panels H, I and J that the anti-nuclear antibodies they observe are almost exclusively kappa light chain positive. 90% of all Igs in mice are kappa positive and in the obvious scenario painted below it would be likely that most responding B cells would be kappa positive.

2. The likely simple explanation of the data is that some IgHlambda2315 immunoglobulin molecule/s which are essentially provided in relatively high doses in these mice, are recognized by a self-reactive BCR or BCRs (at least 60-70% of all BCRs are self-reactive in practical terms) on one or a few B cell clones all of which are kappa expressors (not surprising - there is a 90% chance that a BCR would be kappa positive). Some (possibly a small fraction) of these BCRs in these mice are also capable of recognizing some epitope on a nuclear antigen.

Our response: *The interpretation of the Reviewer is seemingly clever but erroneously based on conventional T-B collaboration (see above). There are several strong arguments against the Reviewer's interpretation.*

- *First, the new $V\lambda 2^{315m}$ mice used herein express $\lambda 2^+$ B cells in physiological amounts, and only low levels of $\lambda 2^+$ Ig (~500 ng/ml, Huszthy et al. PNAS 2019). Thus, the Reviewer's statement: "The likely simple explanation of the data is that some IgHlambda2315 immunoglobulin molecule/s which are essentially provided in relatively high doses in these mice" is not consistent with previous data.*

- *Second, we have extensively tested the specificity of serum autoantibodies (Fig. 1H, I, L, N and SII, lines 104-120). We only found reactivity against nucleosomes and histones and occasional reactivity to dsDNA (Figs. SII and J, lines 118-120). Thus, the autoantibodies in this model are not broadly cross-reactive (e.g to $\lambda 2^{315}$ Ig, like the Reviewer suggests).*

- *Third, and most importantly, we have now tested if four different recombinant lymphoma BCR with specificity for histones/nucleosomes can bind $\lambda 2^{315}$ Ig (myeloma protein M315) in ELISA. None of them did. This speaks strongly against the Reviewer 4's suggestion that autoreactive BCR could cross-react with $\lambda 2^{315}$ Ig, thus initiating conventional T-B collaboration.*

Changes done in the revised manuscript: *The novel data on lack of BCR cross-reactivity to $\lambda 2^{315}$ Ig is included in a Supplementary Fig.S5E and commented upon in the text (Line 272).*

Comment 3.

These kappa+ self-reactive BCRs internalize the IgHlambda2315 immunoglobulin self-antigen, process it and present the lambda peptide to the relatively large numbers of CD4+ T cells that are specific for the peptide. The kappa + self-reactive B cell and the CD4+ T cell collaborate at the T-B boundary generate an extrafollicular B cell response that generates plasmablasts secreting anti-nuclear antibodies. No autoimmune disease necessarily, just an autoimmune phenomenon which is not such a big deal

Our response: *There are several strong arguments against the Reviewer's suggestion, some of which are conclusive.*

- *First, if $\lambda 2^{315}$ Ig is internalized via the BCR, it should be found in endosomes. However, it is not. Rather, the $\lambda 2^{315}$ L chain is localized to the ER (Fig. 3Q and R), which is inconsistent with endocytosis of $\lambda 2^{315}$ Ig.*

- *Second, an ER localization of $\lambda 2^{315}$ was previously found in plasmid-transfected B lymphoma cells, a finding that is consistent with an endogenous origin (Weiss and Bogen Cell 1991). This supports that the ER-localized $\lambda 2^{315}$ in the present lymphomas is endogenously produced.*

- *Third, and most conclusive, we show that lymphoma cells grown in vitro for prolonged periods of time express $\lambda 2^{315}$ in the ER, and display Id peptide on their MHCII molecules (Fig. 3O). Since there is no $\lambda 2^{315}$ Ig in the culture medium, this excludes the reviewer's suggestion of endocytosis of $\lambda 2^{315}$ Ig.*

- *Fourth, and most conclusive, we show that B lymphoma cells express mRNA for $\lambda 2^{315}$ L chains and thus produces $\lambda 2^{315}$ themselves (Fig. 3K).*

Changes to be done in the revised manuscript: *We have added a few words in Discussion to emphasize that $\lambda 2^{315}$ is endogenously produced in the B lymphoma cells. (lines 425-430).*

Comment 4.

As these kappa+ self-reactive B cell clones are repeatedly stimulated by T cell help since antigen is always available, and therefore expand, occasionally additional hits accumulate and lymphomas evolve. The data says most of this!!!

Our response: *As should be clear from the data in the paper and our comments above, κ + self-reactive B cells derive the MHCII-presented Id-peptide not from endocytosed $\lambda 2^{315}$ Ig but rather from endogenously produced $\lambda 2^{31}$. This said, we agree with the latter part of the Reviewer's statement: chronic mutual stimulation between autoreactive, Id-presenting B cells and Id-specific CD4+ T cells results in chronic proliferation, acquisition of mutations, and lymphomagenesis. This idea is already expressed in the Discussion lines 390-397, hence, there should no need for changes to the manuscript.*

Comment 5 (a-d).

The authors should do the following:

- a) use the Wardemann and Nussenzweig approach and clone V regions of Ig H and kappa light chains from some lymphomas, clone into the already widely used Ig vectors, and then express in 293T cells
- b) show that these clonal Igs are self-reactive using the assays they have used in the manuscript

Our response: *We already did the experiment suggested by the Reviewer. We generated recombinant Ig corresponding to BCRs from 4 independent B lymphomas. The following results were obtained:*

- *First, recombinant lymphoma BCRs are indeed specific for histones/nucleosomes, like serum autoantibodies. Importantly, they do not cross-react with $\lambda 2^{315}$ Ig (new experiment, see above). The data are presented in Figure 6 A, B, C and F and Supplementary Figs S4 and S5. The data are commented upon in Results lines 263-290 and in the Discussion lines 398-404.*
- *Second, the recombinant lymphoma BCRs were crucial to demonstrate by proteomic analysis that autoantibodies and lymphoma BCRs share unique sequences. Hence, B lymphomas appear to be derived from autoreactive B cells which is an important conclusion of the paper (Fig. 6 J-L, Results lines 291-3118. Discussion lines 398-404).*

It is somewhat surprising that Reviewer 4 has missed this large body of data so important for the conclusions of the paper. No changes to the paper should be needed.

- c) delete and correct the old schematic in Figure 1

Our response: *As explained above, the Fig. 1A is correct and should be kept.*

Changes done to the manuscript:

As mentioned above, we have modified a sentence in the Introduction, line 42 to emphasize that Id-driven T-B collaboration is distinct from conventional T-B collaboration. We have introduced Fig. 6M and Fig. 7L that further explains the concept of Id-driven T-B collaboration and that the Id peptide can be derived from either the BCR or nascent L-chain in the ER (see also our comments to reviewer 2).

- d) In the abstract and manuscript stick to the use of “autoantibodies” or “autoimmune phenomena” and delete “autoimmunity” in the manuscript. There is no good evidence for an autoimmune disease in this manuscript. We all have autoantibodies! Self-reactivity is a normal phenomenon – some extreme self-reactivity is eliminated by central tolerance mechanisms.

Our response: *We agree with the reviewer that autoimmune disease in mice was mild [slightly reduced gain of weight, inflamed eye lids (blepharitis), moderate hair loss, as*

described in Results lines 106-109 and Fig. 1G, Fig. S1G]. This said, a more fulminant expression of autoimmune disease could have been disguised by lymphoma development.

We agree that the term autoantibodies and autoreactive B cells should be preferred, and we have tried to do so throughout the paper. We have avoided the term autoimmune phenomena since that implies autoimmune disease manifestations, which are mild. However, according to textbook immunology, it is justified to use the term autoimmunity if autoantibodies are detected, which is the case in our model. We do use the term autoimmunity several places in the manuscript and do not wish to change that.

Minor changes to the manuscript (typos and edits)

The authors have made a few changes to the manuscript for reasons of correctness and clarity. None of the changes influence the scientific content of the paper. The changes are marked in the revised manuscript.

Point-to point Rebuttal letter

REVIEWER COMMENTS

Reviewer #1 (Remarks to the Author):

The revised version of the manuscript satisfactorily answered most of the open questions. Only the intranuclear stainings shown in the manuscript require further attention.

1. Please compare the FoxP3 staining shown in Suppl. Fig. S1D to a typical outcome shown on the manufacturer's homepage (clone FJK-16s from Thermo Fisher Scientific; not "FJK-162 by Thermofischer" - see below). What you expect is a clearly separated positive population, not a shift of the entire population and a high background over the whole fluorescence range. To demonstrate specificity of the staining, the authors should plot FoxP3 against CD4 (gated on CD3+ or live) and FoxP3 against TCRb8 or GB113 (gated on CD4+). This would be a better way than the isotype control to judge signal versus background.

***Our Response:** We thank the reviewer for pointing this out. We have now done as the reviewer suggests. We plotted FoxP3 against CD4 (gated on live cells). CD4+ cells were gated for further analysis in a FoxP3 versus TCRVβ8 plot (the transgenic TCR is TCRVβ8.2). The plots are shown in Suppl. Fig. S2D (previously Fig. S1D) where we compare results with Vλ2^{315m+/-} Id-spTCR-TG^{+/-} and Id-spTCR-TG^{+/-} mice by overlays. By this approach a clear population of Vβ8+FoxP3+ CD4 T cells is seen. Based on this staining strategy, we reanalyzed all mice in the group and recalculated data of Fig. 1E so that FoxP3+/splenic CD4+ TCRVβ8+ (%) is now displayed on the y-axis. We did a similar analysis using the clonotype-specific GB113 mAb instead of the anti-Vβ8 mAb. Results were similar but data were clearer with the anti-Vβ8 mAb – which are therefore presented.*

***Changes in the manuscript:** Updated Fig. S2D (previously Fig. S1D) and the corresponding Fig. legend. Updated Fig. 1E and the corresponding Fig. legend.*

2. This manuscript requires careful reading by the senior corresponding author - there are so many inconsistencies and small errors! Please check clone names (see above), spelling of fluorophores (e.g. it is not "PecP-vio700" but "PerCP-Vio700") and manufacturer's names (e.g. it is not "Miltney BioTech" but "Miltenyi Biotec").

***Our Response:** We thank the reviewer for pointing this out. We carefully went through the details in the manuscript and made several minor changes to avoid inconsistencies and minor errors. The fluorophore name PerCP-Vio700 has been updated in Suppl. Fig. S4E and J.*

***Changes in the manuscript:** Updated Suppl. Fig. S4E and J. Several minor changes in manuscript has been made (indicated by track changes, too many to list).*

Reviewer #2 (Remarks to the Author):

My criticism to the first version of the manuscript has been adequately addressed. The manuscript is further improved in response to issues raised by the other referees.

There is one minor point:

In the legend to figure 7, there is a mixup of letters for the subfigures O -Q, and N is missing.

Our Response: *We thank the reviewer for pointing this out. The legend to Figure 7 has been updated and shortened to conform to the length requirement.*

Changes in the manuscript: *Updated legend to Figure 7, lines 1209-38.*

Reviewer #3 (Remarks to the Author):

Our Response: *We thank the reviewer for co- reviewing this manuscript.*

Reviewer #4 (Remarks to the Author):

All issues have been addressed.

Our Response: *We thank the reviewer for reviewing this manuscript.*